# Structural insights into cardiolipin replacement by phosphatidylglycerol in a cardiolipin-lacking yeast respiratory supercomplex

Corey F. Hryc[1], Venkata K. P. S. Mallampalli[2], Evgeniy I. Bovshik[1], Stavros Azinas [1], Guizhen Fan[2], Irina I. Serysheva [2], Genevieve C. Sparagna[3], Matthew L. Baker [2] ✉, Eugenia Mileykovskaya [1] ✉ & William Dowhan [4] ✉

Cardiolipin is a hallmark phospholipid of mitochondrial membranes. Despite established significance of cardiolipin in supporting respiratory supercomplex organization, a mechanistic understanding of this lipid-protein interaction is still lacking. To address the essential role of cardiolipin in supercomplex organization, we report cryo-EM structures of a wild type supercomplex ($IV_1III_2IV_1$) and a supercomplex ($III_2IV_1$) isolated from a cardiolipin-lacking *Saccharomyces cerevisiae* mutant at 3.2-Å and 3.3-Å resolution, respectively, and demonstrate that phosphatidylglycerol in $III_2IV_1$ occupies similar positions as cardiolipin in $IV_1III_2IV_1$. Lipid-protein interactions within these complexes differ, which conceivably underlies the reduced level of $IV_1III_2IV_1$ and high levels of $III_2IV_1$ and free $III_2$ and IV in mutant mitochondria. Here we show that anionic phospholipids interact with positive amino acids and appear to nucleate a phospholipid domain at the interface between the individual complexes, which dampen charge repulsion and further stabilize interaction, respectively, between individual complexes.

The anionic phospholipid cardiolipin (CL) is uniquely localized to mitochondria, where it constitutes approximately 20% of the inner mitochondrial membrane (IMM) phospholipid throughout nature. CL is distinguished from other phospholipids by its unique dimeric structure containing two phosphates, four fatty acids and a free hydroxyl at the 2-position of the glycerol linking the two phosphatidyl groups. This unique dimeric structure makes CL ideal for supporting the organization of individual protein subunits into multi-subunit complexes, as well as individual complexes into supercomplex (SC) molecular machines, such as the mitochondrial respirasome. Crystal structures of the individual respirasome complexes with CL integral to the structures demonstrate its ability to link subunits within individual complexes[1].

The current model of the *Saccharomyces cerevisiae* mitochondrial respiratory chain involves the shuttling of electrons via mobile carriers (coenzyme Q (CoQ) and cytochrome *c* (Cyt *c*)) between either individual free respiratory complexes or individual complexes organized into higher order SCs[2–8]. The *S. cerevisiae* respiratory SC is composed of complexes $III_2$ (CIII, dimer of cytochrome $bc_1$) and IV (CIV, Cyt *c* oxidase) with CIII flanked on each side by CIV in the core tetrameric SC

[1]Department of Biochemistry and Molecular Biology, McGovern Medical School at the University of Texas Health Science Center, Houston, Texas, USA. [2]Department of Biochemistry and Molecular Biology, Structural Biology Imaging Center, McGovern Medical School at the University of Texas Health Science Center, Houston, Texas, USA. [3]Department of Medicine, Division of Cardiology, University of Colorado Anschutz Medical Campus, Aurora, Colorado, USA. [4]Department of Biochemistry and Molecular Biology, Center for Membrane Biology, McGovern Medical School at the University of Texas Health Science Center, Houston, Texas, USA. ✉e-mail: matthew.l.baker@uth.tmc.edu; eugenia.Mileykovskaya@uth.tmc.edu; william.dowhan@uth.tmc.edu

$(IV_1III_2IV_1)$[9–11]. Although there is high sequence homology between yeast and mammalian CIII and CIV, the mammalian core respiratory SC organization differs from yeast in that it contains complex I (CI, NADH:ubiquinone oxidoreductase) and only one CIV[12]. This difference in arrangement is due to asymmetry within CIII where only one of the two monomers contains subunit Sub9. Sub9 in conjunction with assembly factor SCAF1 stabilizes the interaction of CIV over that with CI with one monomer of CIII. The lack of Sub9 in the other monomer allows interaction with CI resulting in CI and CIV flanking CIII.

Our initial discovery of an in vivo requirement for CL for the formation of the SC tetramer $(III_2IV_2)$ in yeast mitochondria[13,14], as well as a requirement for reconstituting the SC from individual complexes in vitro[15] is supported by several reports using cryo-electron microscopy (cryo-EM). CL molecules are present at the interface between CIII and CIV in yeast[10,16,17] and mammalian systems[18,19]. CL may provide a dynamic flexible link between individual complexes within SCs, which allows association/dissociation of individual complexes into/from SC structures that may be required to regulate energy production in response to varying physiological conditions[20,21] as noted below. Reduced CL levels result in a broad range of pathological alterations in mitochondrial function[22] including lower levels of SCs with an increase in free CIII and CIV as observed in neurodegenerative diseases[23], ischemia followed by reperfusion[24,25], induction of apoptosis[26–28], heart failure[29–31], cancer[32], Barth Syndrome[33], hypothyroidism[34], obesity[35], and aging[36]. A presumed advantage in SC formation is bringing active sites of CIII and CIV in close proximity to each other to facilitate Cyt $c$ shuttling of electrons[17,21]. In addition, Cyt $c$ binding to the SC is envisioned to restrict it to 2D diffusion between the above active sites providing a presumed kinetic advantage[21]. In spite of the correlation between the above disease states and reduced CL levels in mammalian systems and compromised respiratory function in yeast mutants lacking CL[13,37,38], there is still considerable debate as to the precise role or advantage SC formation has in cell function.

The requirement for CL in SC formation is generally recognized. Digitonin extracts of wild type (WT) yeast mitochondria when displayed by Blue Native (BN)-PAGE reveals almost exclusively a tetrameric SC $(III_2IV_2)$[13,39]. Digitonin extracts from mutants lacking CL (*CRD1Δ* mutant) when displayed by BN-PAGE showed no higher order SCs and mainly free CIII and CIV[13,39]. However, digitonin extracts of the mutant mitochondria separated by Colorless Native (CN)-PAGE (lacking the blue dye) displayed mostly the tetrameric SC[39]. Supplementation of digitonin extracts of *CRD1Δ* mitochondria with CL restored the display of the tetrameric SC. Therefore, the authors concluded that the SC is present in the mutant mitochondria but is destabilized in the presence of the blue dye used in BN-PAGE. A more recent report[17] indicated that the tetrameric and trimeric (lacking one CIV) SCs are present in WT and mutant mitochondria but the latter extracts showed slightly higher content of the trimeric SC and increased levels of free CIII and CIV. Although there is variability depending on the PAGE system used, lack of CL appears to result in reduced amounts of the tetrameric SC with increased amounts of free CIII and CIV as well as the trimeric SC. This is consistent with our earlier finding of a change in kinetics for Cyt $c$ transfer of electrons between CIII and CIV in mitochondria isolated from *CRD1Δ* cells indicating an increase in free CIII and CIV[14]. This assay was designed to distinguish between 2D diffusion of Cyt c between CIII and CIV in a SC from 3D diffusion of Cyt c between freely diffusing CIII and CIV.

Disruption of the interaction between CIII subunit Cor1 and CIV subunit Cox5a prevented tetrameric and trimeric SC formation in both WT and *CRD1Δ* yeast mitochondria[17]. Since there was little dependence on CL for SC formation, the result suggested that protein-protein interactions are sufficient for SC formation. However, in the yeast *CRD1Δ* cells anionic CL is replaced by a comparable amount of its anionic precursor phosphatidylglycerol (PG)[37], which contains only one phosphate and two fatty acids. Individual complexes with structurally required internal CLs are still functional[14,39], suggesting that PG now replaces CL in individual complexes and possibly in SC formation. To date no high-resolution structural information has been presented for individual respiratory complexes or SCs isolated from yeast mutants lacking CL or from mammalian mitochondria under conditions where CL levels are reduced with increased levels of PG[40] to determine if and where PG might substitute for CL in either individual complexes or SCs.

In this work, we report the cryo-EM maps and molecular models of the WT yeast SC $(III_2IV_2)$ solved to 3.2 Å resolution (Fig. 1 a1 and b1), and the trimeric SC $(III_2IV_1)$ isolated from a *CRD1Δ* mutant solved to 3.3 Å

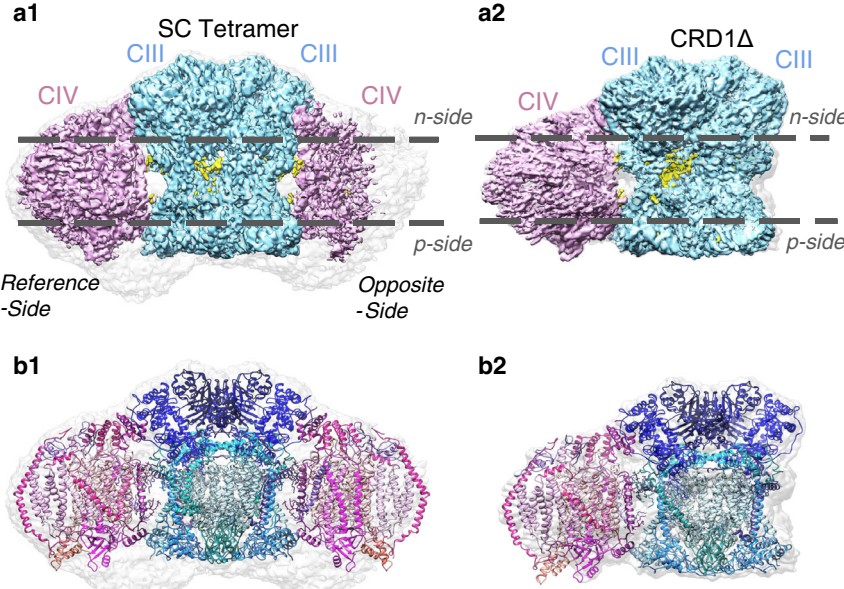

**Fig. 1 | Cryo-EM density maps and models for WT SC (left) and CRD1Δ SC (right).** The segmented cryo-EM density maps for WT SC tetramer (**a1**) and CRD1Δ SC trimer (**a2**) are shown with colors distinguishing CIII (light blue), CIV (pink), ligands, and cofactors (yellow). The grey regions of the density maps show low-threshold density areas, which may be noise, detergent and/or potential structured regions, albeit with flexibility. Models were generated (as discussed in methods) for WT SC tetramer (**b1**) and CRD1Δ SC trimer (**b2**). A ribbon representation of CIII subunits is displayed in shades of blue, and CIV subunits are displayed in shades of pink.

resolution (Fig. 1 a2 and b2), wherein PG replaces CL as the major mitochondrial anionic phospholipid. These two structures provide the basis to study positioning of the anionic phospholipids CL and PG within SCs and for a mechanistic understanding of the requirement for an anionic phospholipid in formation of respiratory SCs.

## Results

### SC purification

WT tetrameric SCs from yeast used for structural analysis have been purified from several strains and by different methods. No previous structure for the trimeric SC isolated from a yeast mutant lacking CL has been reported. We employed sucrose gradient purification of the WT SC[9] and affinity chromatography of Cox4-His-tagged CRD1Δ SC from the mutant *CRD1Δ* strain. Despite different purification methods and strain differences, our WT structure (as discussed below) is in close agreement with previously published structures, which were purified by affinity chromatography of a Qcr7-Flag-tagged[11,17] or Cox13-His-tagged[10,16] SC. More important the protein portion of our structure for the CRD1Δ SC trimer is also in close agreement with our WT SC and the published SCs purified using a tagged subunit of either CIII[11,17] or CIV[10,16]. These identities further validate our approach to comparing the ligand locations between the two structures as we report below.

We verified previous reports that digitonin extracts of WT mitochondria[13] subjected to BN-PAGE displayed predominantly the tetrameric SC with trace amounts of the trimeric SC or free CIII and CIV (See Supplementary Fig. 1). Extracts from *CRD1Δ* mitochondria showed large amounts of the trimeric SC as well as free CIII and CIV with significantly lesser amounts of the tetrameric SC confirming previous results of differences in stability or assembly of the SC in the mutant strain (See Supplementary Fig. 1). WT mitochondria (0.730 ± 0.052 μmoles $O_2$/min/mg, three determinations) showed higher oxygen consumption than the *CRD1Δ* mitochondria (0.360 ± 0.047 μmoles $O_2$/min/mg, three determinations) upon addition of NADH (See Supplementary Fig. 2 and Supplementary Table 3). Previously we reported that oxygen consumption was about the same for the WT and ΔCRD1 mitochondria (0.32-0.35 μmoles $O_2$/min/mg) isolated from strains with a different genetic background[14]. The determination of oxygen consumption for digitonin extracts of mitochondria is not a meaningful comparison of coupled transfer of electrons within the tetrameric and trimeric SCs. WT activity is a direct measure of coupled electron transfer between CIII and CIV since the extracts contain almost exclusively the tetramer. However, given the large amount of free CIII and CIV in the mutant extract, the mutant activity is a composite of electron transfer within the trimer and between freely diffusing CIII and CIV. The purified WT SC supplemented with Cyt *c* and reduced decylubiquinone (DQH2) also displayed similar activity (1.86 ± 0.07 μmoles $O_2$/min/mg, three determinations) to that (1.0 μmoles $O_2$/min/mg) previously reported[9] (See Supplementary Fig. 2 and Supplementary Table 3). However, the purified trimeric CRD1Δ SC displayed much lower activity (0.350 μmoles $O_2$/min/mg, one determination) consistent with low in-gel activity for CIII (See Supplementary Fig. 3). The purified trimeric CRD1Δ SC lacked CIII subunit Qcr10 (see below), which was also reported largely missing when CIII was purified for crystallographic analysis[41]. CIII purified from a Qcr10-lacking strain showed a 40% reduction in activity[42], which, coupled with the lower activity of ΔCRD1 mitochondria, most likely contributes to the low activity for the purified ΔCRD1 SC.

### Structure of the WT yeast III2IV2 SC

The cryo-EM map displayed a global resolution of 3.2 Å (for data collection and image processing see Methods, Supplementary Tables 1 and 2, and Supplementary Figs. 4 and 5) and revealed CIII flanked by CIV monomers on each side (Fig. 1 a1), which is in line with previously published structures[10,11,16,21]. Local resolution of the final density map shows that the central part of the map, which includes CIII, is better

resolved than the CIV monomers (Supplementary Fig. 5f), consistent with the suggested dynamic character of CIV[10,11,16,21]. The reconstruction also revealed that one CIV was better resolved as previously shown[11]. We denoted this side as the reference-side, with the weaker resolved side being denoted as the opposite-side. Our subsequent findings focus on the reference-side of the map, unless otherwise noted. For construction of the WT SC tetramer model IV1III2IV1 (Fig. 1 b1; Supplementary Fig. 6), we used the previously reported structure of the SC from *S. cerevisiae* (pdb: 6HU9)[10]. While the 6HU9 model generally fit the density, additional rigid-body adjustments and local refinements were necessary. The adjusted model was then refined with ligand restraints in Phenix[43]. See the detailed description of the model building in Methods.

Although our model of the WT SC (Fig. 1 b1 and Supplementary Fig. 6) is in close agreement with published models, examination of our map revealed differences with the published results. Specifically, a difference map revealed discrete densities within CIII and at the interface of CIII and CIV in our reconstruction that were not seen in the previously published structure (pdb ID 6HU9)[10]. To model these unknown densities, previously resolved respiratory SCs and CIII (pdb ID: 6YMX[17], 1KB9[44], 3CX5[45], 6Q9E[18], 6GIQ[11]) were fit in the map. We identified an extra density located at the UQ6 (ubiquinone) position of 6YMX (C406) ($Q_i$-inside site; also known as $Q_N$ negative site). Further analysis of the difference map revealed strong excess density at the location of SMA (stigmatelin) seen in 1KB9; however, our sample preparation did not include this inhibitor. This density was observed on both the reference-side and opposite-side of the complex. UQ6 (UQ6 606 J, UQ6 606j) was modeled at those locations ($Q_o$-outside sites, also known as $Q_P$ -positive sites). Since at the reference-side UQ6 (606 J) lacked strong tail density, the model was truncated to match the density. In total, four UQ6 ligands were placed into the map, two in each monomer of CIII, (UQ6 603 J, UQ6 603j at $Q_i$ sites and UQ6 606 J, UQ6 606j at $Q_o$ sites) with one UQ6 (606 J) having a large break in the tail density (Fig. 2). All previously reported structures of the *S. cerevisiae* respiratory SC contained UQ6 bound in the $Q_i$ sites but empty $Q_o$ sites[10,11,21]. Occupancy of both $Q_o$ sites is favored by reduced UQ6, which may be due to a gentler method of purification as compared to the chromatography methods used by others. Further studies of the SC purified by sucrose gradient centrifugation and its reduction/oxidation state are required to provide a detailed explanation of this arrangement.

In our structure, UQ6 interacts with His181 of the head domain of Rip1 (Rieske protein) in the $Q_o$ site of the opposite-side monomer of CIII, which interacts with the more poorly resolved CIV monomer. The $Q_o$ site on the reference-side of CIII, which interacts with the reference CIV, lacks a strong UQ6 tail density, which might be an indication of increased UQ6 flexibility at this site. The position of the Rip1 flexible domain on both sides is rather similar to its position in the 3CX5 crystal structure of CIII (*b* position)[45] than to positions of this domain in the 6HU9 or 6YMX SC structures (intermediate position, between *b* and *c* positions[10,17] Supplementary Fig. 7).

Finally, our density map at the 2-fold of CIII does not accommodate UQ6, which was modeled at this position in 6HU9. Various attempts to fit this density were made based on previously known SCs. CN5 (CL) from 6YMX was the best fit for this density, thus, it was modeled at this site (Supplementary Fig. 6). Subunit/chain names and positions for the WT SC are summarized in Supplementary Table 4 and Supplementary Fig. 8a, respectively.

### Localization of CL in the WT SC structure

Non-protein densities in our WT SC structure were all observed at the locations of CL (Fig. 3a, CDL = CL) in the previously solved cryo-EM structures of the yeast SC[10,11,16,21]. Thus, we assigned these densities to CL molecules (see Methods). The hydrophilic headgroups of these CLs interact with amino acid side chains through hydrogen bonding and/or

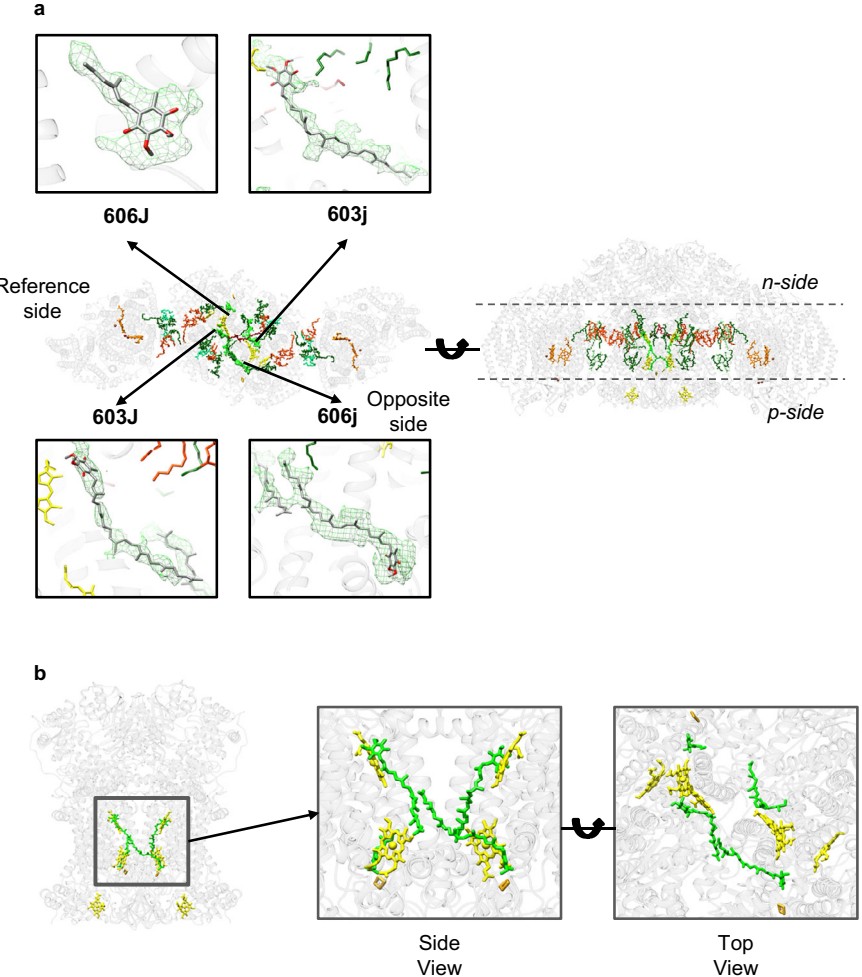

**Fig. 2 | Gallery of UQ6 ligands for the WT SC. a** The model is shown with isolated density for each of the four UQ6s identified in the density map. The truncated UQ6 606 J is on the p-side of the reference side of CIII, which interacts with the reference side of CIV. **b** UQ6 (shown in green atom representation) position is highlighted with respect to CIII (grey). In addition to FeS (gold) and Hemes $b_H$ and $b_L$ (yellow) are shown.

salt bridges (Fig. 3b). Two CL molecules (CDL601K/k and CDL302C/c) at each CIII/CIV interface and one CL (CDL501J/j) within each of the monomers of CIII were fit into these densities (Fig. 3a) allowing us to assign possible interactions of the CLs with the SC proteins and other phospholipids. One phosphate residue of CDL501J (Fig. 3b and Supplementary Fig. 9a) is in close proximity to Arg4 of Cob and His345 of Cor1 within CIII. Only CDL601K (Fig. 3b and Supplementary Fig. 10b) appears to interact with both CIII and CIV through phosphate moieties and Lys35 of Qcr8 in CIII and Lys487 of Cox1 in CIV, respectively. CDL302C (Fig. 3b) is in close proximity to CDL601K, and its phosphates appear to interact with Lys44 and Lys51 of Rip1 on the surface of CIII at the interface with CIV (Supplementary Fig. 10a). A density for CDL402L (Fig. 3a and b) in which one phosphate is in close association with Lys288 of Cyt1 and Tyr28 of Cob in CIII was also observed (Supplementary Fig. 11a). CDL601H (Supplementary Fig. 11b) fit a density also in association with CIII in which one phosphate appears to interact with His85 in Qcr7, and the other phosphate appears to interact with Arg8 of Qcr8. Interestingly, CDL302C is in close proximity to both CDL601K and CDL402L with the latter in close proximity to CDL601H (Supplementary Figs. 10 and 11. These CLs appear to form a hydrophobic domain that may enhance binding between CIII and CIV by stabilizing CL at the interface. CDL402L like CN3 (CL), previously identified by X-ray crystallography[1,44], lies in a groove of CIII and neutralizes positively charged residues of subunit Cyt1 near CDL402L. Although this area of the CIII surface where CDL402L and CDL601H are

located is near the CIII/CIV interface, it does not directly face the surface of CIV.

## Structure of the III₂IV₁ trimeric SC isolated from the *CRD1Δ* mutant

To address whether PG substitutes for CL in SC formation, we performed single-particle cryo-EM analysis of the trimeric SC purified from *CRD1Δ* cells. Using mass spectral analysis (Supplementary Fig. 12 and Supplementary Table 5), we confirmed that the mitoplasts (and therefore the IMM) isolated from the *CRD1Δ* mutant contains the immediate precursor of CL, phosphatidylglycerol (PG) and lacks CL as previously reported for whole yeast based on radiochemical labeling[37,38]. In *PGS1Δ* mutants that lack CL and PG, the mitochondrially encoded components of CIII (Cob) and CIV (Cox1, Cox2 and Cox3) are no longer synthesized[46]. Therefore, no definitive structural requirement for PG in place of CL can be concluded from genetic manipulation. To date, the possible replacement of CL with PG in individual respiratory complexes and SCs has been suggested but not documented. Grid preparation and cryo-EM data collection were the same as for the WT. Due to the preferred orientation, additional images were collected at 30° tilt. Image processing and final refinement using a tight mask around the SC resulted in 3.3 Å density map (Fig. 1 b2). For details see Methods, Supplementary Tables 1 and 2, and Supplementary Figs. 13 and 14.

To construct a model of the CRD1Δ SC optimized protein subunits from CIII and CIV were extracted from the WT model and fitted

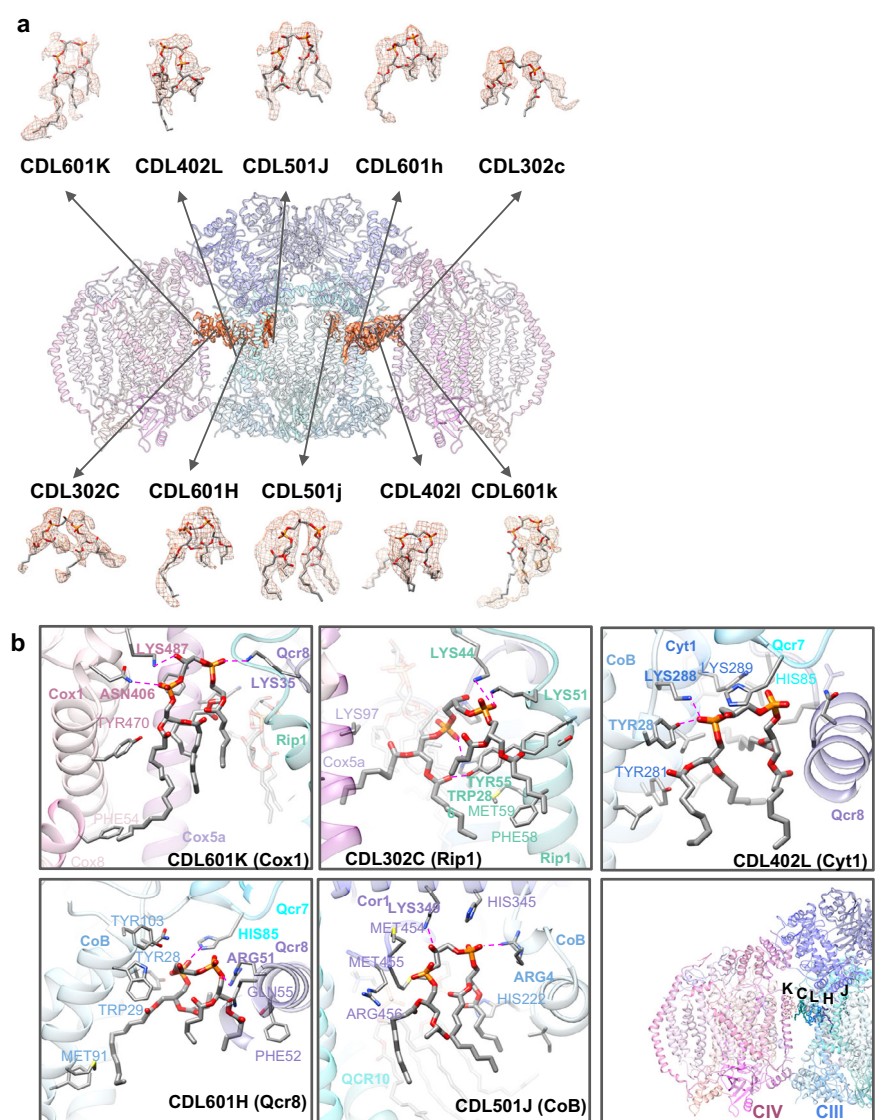

**Fig. 3 | Gallery of CDL (CL) ligands in the WT SC structure. a** The model is shown with isolated density for each of the CDL ligands identified in the density map. A zone equivalent to the resolution (3.2 Å) was used to isolate the density map corresponding to the model. **b** CDL potential interactions with neighboring subunits of the SC tetramer. The five CDLs are shown on the reference-side of the WT SC tetramer. Side chains within 4 Å of the focused CDL are displayed, with specific interactions being highlighted with bold side chain labels and a magenta dashed line. CDLs are labeled according to the chain/subunit names, see Supplementary Table 4.

independently into the CRD1Δ trimeric SC 3.3 Å density map with Chimera, followed by refinement in Phenix and COOT. There was no density in the region of the CRD1Δ density map corresponding to the position of the Qcr10 subunit of CIII in the WT SC density map; we assume that this subunit was lost during purification. All other subunits of CIII and CIV in the mutant showed a similar organization and location as in the WT structure (Supplementary Figs. 8b, 15 and 16). Densities for UQ6 were found only at Qi sites (Supplementary Fig. 15). The $Q_o$ sites were empty, which may be due to use of chromatography for purification resulting in the oxidation of UQ6. Analysis of the cryo-EM density for Rip1 subunit of CIII showed that both the hinge region and the head domain displayed weaker or less ordered density when compared to the WT. This variation in density may indicate that the head domain of Rip1 in the CRD1Δ CIII is flexible and exists in a larger conformation space. This is consistent with the finding that this ectodomain is stochastic when $Q_o$ site is empty[47]. In addition, there was no density at the position similar to CN5 (CL) in the WT structure. Importantly, the mutual orientation of the individual CIII and CIV from the *CRD1Δ* mutant was the same as in the WT SC (Fig. 1 b2 and a2,

Figs. 4a and 3a). This allowed us to make a comparative analysis of the lipid positions in the mutant and WT structure and focus on areas which in the WT SC were occupied by CL, including the interface between CIII and CIV. Subunit/chain names and positions for the CRD1Δ SC are summarized in Supplementary Table 6 and Supplementary Fig. 8b, respectively.

**Localization of phosphatidylglycerol (PG/PGT) in the CRD1Δ SC**
As described in Methods, the CRD1Δ map was masked using the protein model to reveal unmodelled densities for cofactors and lipids. Hemes, appropriate ligands (PEF (PE), PCF (PC)) and cofactors were then extracted from the WT map and rigid body fit to the unmodeled parts of the CRD1Δ density map. Importantly, the positions corresponding to the CDL locations in the WT SC did have densities indicating the presence of ligands. The density at these positions revealed two tails with one strong headgroup as opposed to the four tails and two strong headgroups for CDL (Fig. 4a, b). This correlated with positioning of the anionic phospholipid PGT (PG), which fit into the CDL positions. To further test and confirm the fitting of PGT to these

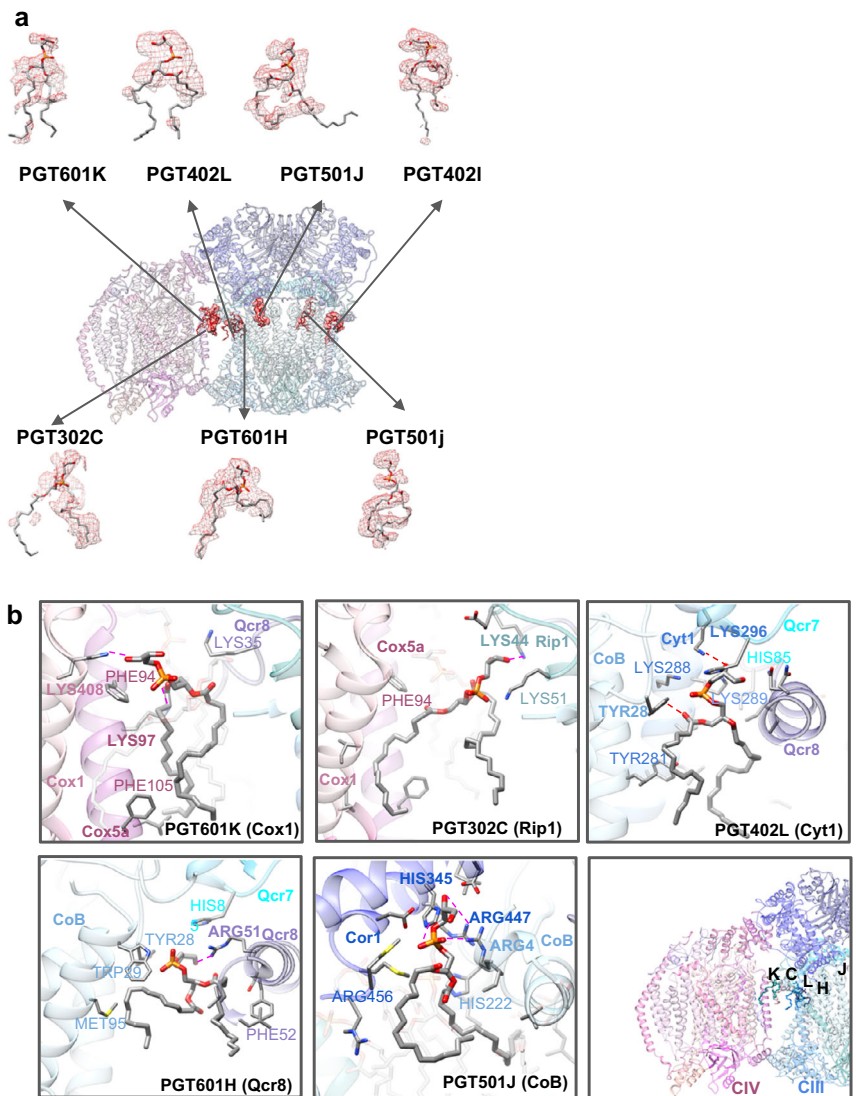

**Fig. 4 | Gallery of PGT (PG ligands in the CRD1Δ SC structure. a** The model is shown with isolated density for each of the PGT ligands identified in the density map. A zone equivalent to the resolution (3.3 Å) was used to isolate the density map corresponding to the model. **b** PGT potential interactions with neighboring sub-units of the CRD1Δ SC. The five PGT ligands are shown on reference-side of the CRD1Δ SC. Side chains within 4 Å of the focused PGT are displayed, with specific interactions being highlighted with bold side chain labels and a magenta dashed line. PGT are labeled according to the subunit/chain names (see Supplementary Table 6).

densities, PEF, PCF, and CDL were all fit into the suggested PGT locations. One iteration of refinement was completed with the ligands in the PGT density. Next, a calculated density map was computed for each of the ligand groups and the threshold was set to be equivalent for each ligand group. Finally, the cross-correlation score was computed for the six sites. PGT resulted in the highest average correlation (Supplementary Table 7).

PGT501J/j (Supplementary Fig. 9b) appears to replace the internal CLs in each monomer of CIII surprisingly resulting in little noticeable structural changes for CIII. The one phosphate of this PG is near Arg4 of Cob as seen for CDL in this position. The remaining PGTs also interact with similar regions of the SC as CDLs but with fewer or weaker interactions. The headgroup of PGT302C (Supplementary Fig. 10c) lies near Lys44 of Rip1 (as does CDL302C) and may interact with a hydroxyl of its glycerol headgroup rather than through the phosphate. However, there is no interaction with Lys51 of Rip1 since PGT lacks a second phosphate. PGT601K (Supplementary Fig. 10d) appears to only interact with CIV unlike CDL601K which interacts with both CIII and CIV. Rather than stronger potential ionic interactions as observed for CDL601K, only weaker interactions between a hydroxyl of the glycerol headgroup and Lys408 in Cox1 of CIV and the carbonyl of the *sn*−1 fatty acid of PGT and Lys97 in Cox5a of CIV were observed. The one phosphate of PGT402L (Supplementary Fig. 11c) is in close association with Lys296 of Cyt1 and Tyr28 of Cob in CIII. PGT601H (Supplementary Fig. 11d) is in close association with Arg51 of Qcr8 in CIII but lacking a second phosphate does not interact with His85 in Qcr7 of CIII. PGT601H and PGT402L like PGT302C and PGT601K are near each other, but unlike in the case of CL, PGT302C and PGT402L are close (Fig. 4a) but appear to be further apart than CLs at this site (Supplementary Figs. 10 and 11).

## Discussion

The primary goal of this work was to understand the role of lipids in the organization of respiratory SCs. SCs from yeast have been purified by a variety of methods from different strains yet the reported structures, including our structures, are remarkably similar. Although the purification methods used to obtain the WT and CRD1Δ SCs were different, the structures were very similar except for the lipid and UQ6 components, which supports the validity of comparing structural results with respect to the non-protein ligands. Our structural

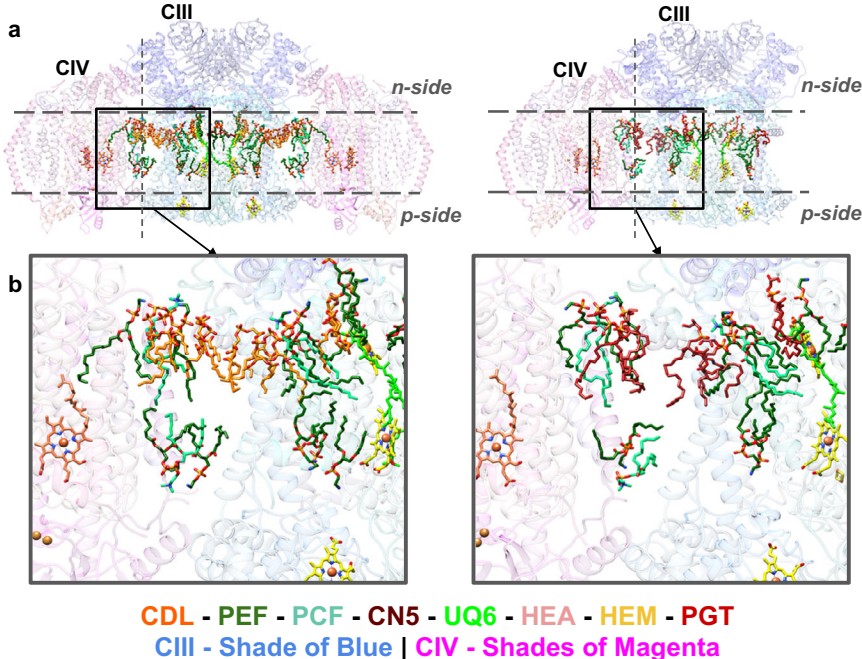

**CDL - PEF - PCF - CN5 - UQ6 - HEA - HEM - PGT**
**CIII - Shade of Blue | CIV - Shades of Magenta**

**Fig. 5 | Location of identified lipids and cofactors. a** An overview of the WT SC tetramer (left and CRD1Δ SC trimer (right) with lipids and cofactors colored by type. **b** A zoomed-in region reveals the lipid arrangement at the CIII/CIV interface; WT SC (left) and CRD1Δ SC (right).

determination of the trimeric SC from *CRD1Δ* cells shows the presence of the anionic phospholipid PG at all CL positions found in the WT SC (Fig. 4), which has been postulated but never demonstrated. However, there were significant differences as to which amino acids interact with PG and none of the PGs formed interactions that bridge CIII and CIV. These differences in the interaction of the individual complexes with the two anionic phospholipids may explain the reduced detection of the tetrameric SC and the presence of the trimeric SC and individual CIII and CIV when analyzed by BN-PAGE. The presence of significant amounts of free CIV and CIII in the absence of CL in vivo is sufficient to explain alterations in the kinetics of Cyt *c* transfer of electrons between CIII/CIV as we previously demonstrated when comparing mitochondria isolated from WT and *CRD1Δ* cells[14].

An additional lipid stabilizing factor for SC formation appears to be anionic lipid nucleation of lipid domains at the interface of CIII/CIV. The fatty acid hydrophobic tails of all the CLs are near side chains of hydrophobic amino acids and other phospholipids (PE and PC), which further stabilizes these interactions. Although CDL402L and CDL601H appear not to lie within the interface between CIV and CIII, the former is near CDL302C (Supplementary Figs. 10d and 11c), which in turn is in close proximity to CDL601K (Supplementary Fig. 10c, d). CDL601K is the only CL that interacts with both CIII and CIV. Furthermore, additional phospholipids (PE and PC) (Fig. 5a, b, left) are resolved in close association with the above CLs at or close to the interface. Additional phospholipids most likely fill the empty spaces between CIII and CIV but might be too flexible to be resolved in the structure. Given that reconstitution of the tetrameric SC from individual complexes in vitro only occurs when CL is present[15] strongly suggests that CL nucleates a domain of phospholipids at the CIII/CIV interface dependent on specific interactions of CL with SC protein components, thus further stabilizing SC formation. As with CLs, the fatty acid hydrophobic tails of all the PGs are near side chains of hydrophobic amino acids and other phospholipids (PE and PC) (Fig. 5a, b, right), which may also form a hydrophobic domain that further stabilizes the trimeric SC. However, the hydrophobic space occupied by PG is considerably less than that of CL, which in turn lowers its potential to stabilize hydrophobic interactions. As shown in the electrostatic potential map (Fig. 6) of the WT

tetrameric and CRD1Δ trimeric SCs, the negatively charged headgroups of CL and PG, respectively, are positioned in close association with a positively charged surface at the CIII/CIV interface. This neutralization of positive charges may allow close association of CIII and CIV and further supports a requirement for an anionic lipid in SC formation. In addition, more global hydrophobic interactions of CL molecules or PG molecules with other phospholipids and amino acid side chains are also significant contributors to the formation of SCs (Fig. 5). Therefore, anionic phospholipids (our conclusion) in conjunction with protein-protein interactions[17] are primary stabilizing forces for SC formation with CL possibly being more stabilizing than PG.

## Methods
### Yeast strains and cell growth
*S. cerevisiae* strain USY00b (Mat*a, ade2-1, his3-11-15, leu2-3-112, trp1-1, ura3-52, can1R-100, atp2::LEU2, TRP1::ATP2-His₆*) expresses the $F_OF_1$-ATPase subunit 2 with a C-terminal $HIS_6$ extension[48]. Lipid extracts of mitoplasts contained low levels of PG compared to the expected levels of CL (Supplementary Fig. 12 and Supplementary Table 5). The strain was cultivated in a growth medium containing 1% yeast extract, 2% peptone, 3.7% of lactic acid or 1% ethanol (v/v), and 3% glycerol at 30 °C. The cells were harvested ($OD_{600}$ of 2.0-3.0) by centrifugation at 4,700 x g and washed with cold TBS buffer (50 mM Tris-HCl, pH 7.4 containing 150 mM NaCl).

A *CRD1Δ* strain (named YEB100) that encodes the Cox4p subunit of CIV with a 10-residue His-tag at its C-terminus was constructed as follows. Plasmid pFA6a-link-yomTagBFP2-Kan (Addgene, catalog # 44899) was used as a template to amplify the KanMX kanamycin-resistant cassette under TEF promotor/terminator control. 5' to 3' forward and reverse primers CAGGCCTGGTAGCATAGTTTGGTCCC TAATAATTTAGTCAATGGACATGGAGGCCCAGAAACCCTC and AAAA GTCAGGACCCTTTTCAAAAAGGATCGCAATTATACTACAGTATAGCGA CCAGCATTCACATACGATTGA, respectively, were designed to isolate the KanMX cassette flanked by upstream and downstream sequences at the *CRD1* locus. The DNA isolate was used to transform a previously constructed strain carrying the *COX4-His₁₀* gene (Mat*a his3 leu2 ura3*

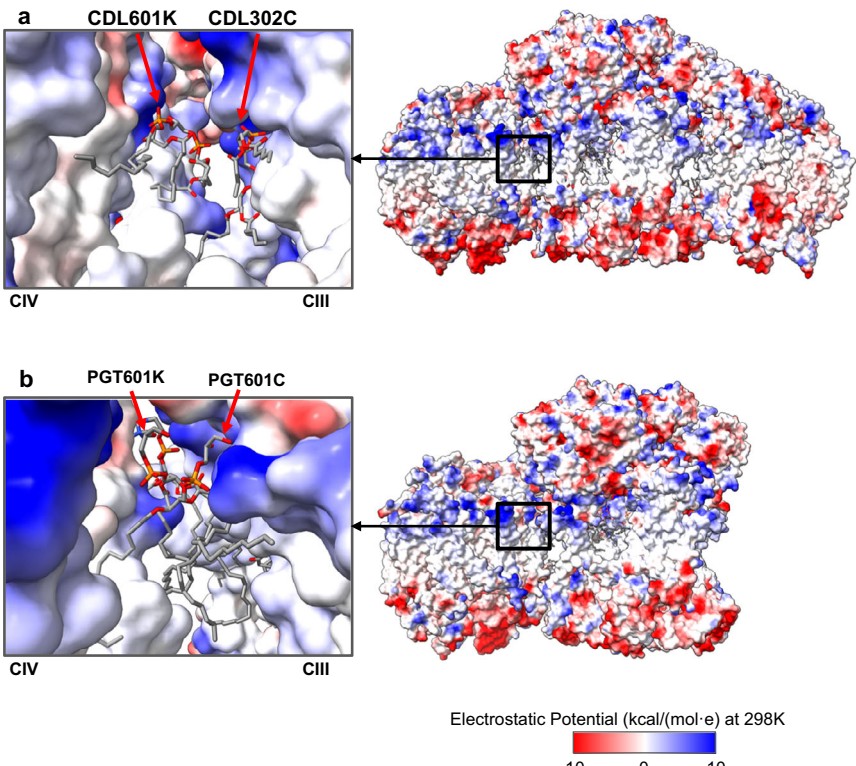

**Fig. 6 | Electrostatic potential map of the WT SC and CRD1Δ SC.** Electrostatic potential map (generated in Chimera× 50 based on Coulomb's law) is mapped onto the WT SC (**a**) and CRD1Δ SC (**b**); (blue, positive charge and red, negative charge). The boxed-out region (on the left) reveals the electrostatic pocket at the CIV-CIII interface. The headgroups of CDL x 2 (**a**) and PGT × 2 (**b**) have complementary charges when compared to the interacting surface of the protein.

*COX4-His*[10][15]. Transformants were selected for growth on 1% yeast extract, 2% peptone, and 2% dextrose agar plates supplemented with 200 μg/ml G-418 (Gibco Geneticin Selective Antibiotic, # 10131-035). The resulting strain (Mat*a his3 leu2 ura3 CRD1::KanMX4 COX4-His*[10]) was verified by PCR analysis for the replacement of the *CRD1* gene by the *KanMX4* cassette. Lipid extracts of mitoplasts lacked detectible CL and contained high levels of PG (Supplementary Fig. 12 and Supplementary Table 5). The strain was grown in 1% yeast extract, 2% peptone, 1% ethanol (v/v), and 3% glycerol (v/v) at 30 °C. Cells were harvested (OD$_{600}$ of 3.0) by centrifugation at 4,700 x g and washed with cold PBS buffer (8.0 g/L NaCl, 0.2 g/L KCl, 1.42 g/L Na$_2$HPO$_4$, 0.24 g/L KH$_2$PO$_4$, pH 7.4).

## Isolation of mitochondria

Mitochondria from WT and *CRD1Δ* strains were isolated from spheroplasts of yeast cells[14]. Cells were harvested (OD$_{600}$ of 2 to 3) by centrifugation at 4700 x g for 10 min at 4 °C and washed with ice-cold 1X TBS. The cell pellet was then resuspended in 100 mM TrisSO$_4$ (pH 9.4) containing 10 mM dithiothreitol at a ratio of 2 ml/g weight of cell pellet and incubated at 30 °C for 20 min with shaking, centrifuged at 750 x g at 4 °C for 10 min and then washed with 2 ml of 1 M sorbitol per gram of cell pellet. The yeast cell wall was digested by incubating the cells with zymolase-20T (MP Biomedicals, # 320921) at 3 mg/g weight of cells in buffer containing 1 M sorbitol and 20 mM potassium phosphate buffer (pH 7.5) for 90 min at 37 °C with constant shaking. The resulting spheroplasts were washed with 1 M Sorbitol (2 ml/g of cells) at 4 °C and disrupted with a Dounce homogenizer in buffer containing 0.5 M sorbitol, 10 mM Tris-HCl (pH 7.2), 0.02% bovine serum albumin, 1/1000 volume of protease inhibitor cocktail set III (Calbiochem Millipore, # 539134)) and 1 mM phenylmethylsulfonyl fluoride. The lysed cells were centrifuged at 1700 x g at 4 °C for 5 min, and the supernatant was centrifuged at 8500 x g at 4 °C for 20 min. The pellet was suspended in 1 ml of SEM buffer (250 mM sucrose, 10 mM Mops (pH 7.2), 1 mM EDTA) and loaded onto a 15–60% sucrose gradient in 15 mM Tris-HCl (pH 7.4) plus 20 mM KCl that was centrifuged at 111,000 x g at 4 °C for 90 min. The mitochondrial layer was aspirated, flash frozen with liquid nitrogen and stored at −80 °C. Protein concentration of the mitochondria was measured using the BCA (Pierce Biotechnology, # 23227) method with bovine serum albumin as standard according to the manufacturer's directions.

## Respiratory activity assays

The Oxygraph-2k high resolution respirometry system (Oroboros Instrument) was used to measure oxygen consumption rates of the WT and the *CRD1Δ* mutant purified mitochondria and purified SCs (See Supplementary Fig. 2 and Supplementary Table 3). Measurements were performed at 30 °C. The buffer used contained 0.5 mM EGTA, 3 mM MgCl$_2$, 60 mM lactobionic acid, 20 mM taurine, 10 mM KH$_2$PO$_4$ at pH 7.1, 20 mM HEPES, 110 mM sucrose in a total volume of 2 ml. The reaction was started by the addition of 0.5 mM NADH and stopped by the addition of 2.5 μM antimycin A. For mitochondrial measurements, one sample was measured three times to arrive at an average activity with a standard deviation. In the case of the purified SCs incubation media was supplemented with 0.05% digitonin (w/v) and purified Cyt *c* (50 μM). The reaction was started by the addition of NaBH$_4$-reduced decylubiquinone (Sigma, # D7911) (DQH$_2$ 100 μM) and stopped by the addition of 2.5 μM antimycin A. Due to limited amounts of the ΔCRD1 SC, only one determination was possible. The chamber volume was 2 ml, and protein concentration was recorded in mg/ml as noted in Supplementary Fig. 2 and Supplementary Table 3. The software package associated with the instrument is DatLab-version 7.3.0.3. Oxygen concentration was monitored in μmolar concentration and converted to μmoles of total oxygen consumed per total mg of protein as reported in the text.

## ESI-MS analysis of CL and PG

Presence of PG in place of CL in the *CRD1Δ* strain IMM (Supplementary Fig. 12 and Supplementary Table 5) was confirmed by CL and PG molecular species quantification in mitoplast samples by liquid chromatography (LC) coupled to electrospray ionization mass spectrometry (MS)[31]. Mitoplasts were prepared from mitochondria suspended in SEM buffer by dilution 15-fold with 20 mM HEPES-KOH (pH 7.4). After incubating on ice for 30 min, the mitoplasts were collected by centrifuged at 14,000 x *g* for 10 min at 4 °C and redissolved in SEM buffer. Sample preparation details are as follows: Lipids from 250 µg of mitoplasts protein (quantified using Pierce BCA protein assay) from WT and *CRD1Δ* strains were extracted by adding 800 µl methanol, 1 µmol tetramyristoyl-CL as an internal standard (Avanti Polar Lipids, Alabaster, AL), and 715 µl acidified salt solution (0.1 N HCl + 11 mM ammonium-SO4). 1 ml chloroform was added to induce phase separation, and samples were vortexed then centrifuged 5 min at 2500 x *g*. The bottom layer was then transferred to a clean tube and dried under a stream of nitrogen. The dried lipids were resuspended in 100 µl hexane:isopropanol (30:40) and transferred to an autosampler collection tube. To determine the amount of CL and PG in each sample, a standard curve with four reference standard amounts of 0.34, 0.68, 1.36, and 2.72 nmol of both $(18:1)_4$ CL and 16:1 18:2 PG (both from Avanti Polar Lipids, Alabaster, AL) was made and extracted as detailed above. 25 µl of each sample or standard was injected (Shimadzu LC system model LC-20AD) using binary flow LC with a flow rate of 0.2 ml/min using normal phase solvents A: hexane:isopropanol (30:40) and B: hexane:isopropanol:10 mM ammonium acetate (30:40:8.4). Percent B started at 45%, then 50% at 10 min, 98% at 10.5 min and 45% at 25 min. The run time was 32 min. The mass spectrometer used was an API 4000 QTRAP (Sciex, Framingham, MA) run in negative polarity mode with samples collected in Q3 with a full scan range from m/z 600 to 1700. Settings on the mass spectrometer were curtain gas 20 psi, ion spray voltage −4500 V, temperature 300 °C, nebulizer gas 20 psi, heated gas 30 psi, interface heater ON, declustering potential −35 V, entrance potential −10 V, collision cell exit potential −12 V, and electron multiplier of the detector 2250. Analysis was done with Analyst version 1.6.2 software. For PG and singly ionized CL peaks, area under the time curve was used for the peak ± 0.5 mass units. Background values from a blank with only solvent was subtracted from these counts. The counts were then converted to nmol/mg protein using the slopes of the respective standard curves for either CL or PG. CL and PG were identified using their retention times obtained from the retention time of the reference standard compounds in the standard curves and their molecular masses. Additionally, singly ionized CL peaks were confirmed using the presence of a matching doubly ionized peak. The doubly ionized CL peak is distinctive because the isotope peaks are separated by a half mass unit in contrast to singly ionized PG and other two fatty acyl phospholipids where their isotope peaks are separated by one mass unit. One measurement was made for each lipid extract, so no statistics are provided.

## Purification of SCs

The $III_2IV_2$ SC from the CL-containing WT USY00b strain was purified by the method we previously developed[9]. Isolated mitochondria (8 mg of protein) were suspended in the 1 ml of lysis buffer containing 2% (w/v) digitonin (Sigma, # D141-500MG), 50 mM potassium acetate, 10% glycerol, 1:50 volume protease inhibitor cocktail set III (Calbiochem Millipore, # 539134), 1.5 mM phenylmethylsulfonyl fluoride and 30 mM HEPES-KOH, (pH 7.4) for 1 h at 4 °C with gentle shaking. After incubation the lysate was centrifuged at 4 °C for 20 min at 90,700 x *g*. 1 ml of supernatant was incubated with 0.45 ml of magnetized Cobalt Beads (Dynabeads TALON catalog #101.02D, Invitrogen) for 45 min at 4 °C with constant shaking to remove $F_oF_1$-ATPase. Prior to use, beads were washed 3 times with TBS using a magnetic separator (Magna-Sep™ Magnetic Particle Separator, Invitrogen) to recover the beads. After

incubation the beads were removed using a magnetic separator. The supernatant (1 ml) was immediately layered onto an 8 ml sucrose gradient (0.75 M to 1.5 M sucrose in 15 mM Tris-HCl (pH 7.2), 20 mM KCl, and 0.4% digitonin) and centrifuged at 4 °C for 20 h at 111,000 x *g*. Fractions (80–100 µL) from the gradient were analyzed by BN-PAGE as described in the next section. Selected fractions containing the purified $III_2IV_2$ SC were combined and used for cryo-EM. Protein concentrations were determined using the BCA protein assay kit (Thermo Scientific) according to manufacturer's instructions.

The $III_2IV_1$ trimeric SC from mitochondria of CL-lacking (*CRD1Δ*) YEB100 strain was purified as follows. Mitochondria were solubilized as described above. The clarified lysate was loaded at 1 ml/min onto a 5 ml His-Trap HP column (Sigma, cat. # GE17-5248-02) that was previously equilibrated with 50 mM HEPES, 500 mM NaCl and 0.05% digitonin, pH 8.0 (Buffer B). The loaded column was washed with several volumes of Buffer B. Proteins were eluted with 20% of Buffer B containing 100 mM Imidazole. Fractions containing the highest absorbance at 280 nm were subjected to size exclusion chromatography to separate the $III_2IV_1$ SC from contaminating individual CIII and CIV. A Superose 6 Increase 10/300 GL column (GE Health Care, cat. #29-0915-96) was pre-equilibrated with column buffer containing 50 mM Tris-HCl (pH 7.2), 150 mM NaCl, 0.05% digitonin. A sample (2 ml) from the His-Trap column was loaded and eluted with column buffer. Fractions were collected based on peak intensity at 280 nm. All fractions were analyzed by BN-PAGE. Fractions containing the $III_2IV_1$ SC and free of CIII and CIV were pooled and used for cryo-EM.

## BN-PAGE, CN-PAGE, Western blot analysis, and in-gel enzyme activity

Pre-made 3–12% acrylamide gels containing 5% Coomassie Blue G-250 (Invitrogen, cat. # BN2008)[15] were used for BN-PAGE. Running times were 14 h at an initial current of 12 mA at 4 °C. After gels were stained with Bio-Safe Coomassie Blue G-250 (BIO-RAD, cat. # 1610786), proteins were electroblotted to a PVDF membrane (Invitrogen, cat. # LC2005) for Western blot analysis. For CIII or CIV visualization custom polyclonal antibodies (prepared by Cocalico Biologicals) against purified CIII (raised in rabbits, 0.46 mg/m, diluted 1:2,500) and purified CIV (raised in guinea pigs, 0.77 mg/m/ diluted 1:5,000) were used. These antibodies were verified by detection of purified CIII and CIV as well as only complexes containing CIII or CIV in crude extracts[15]. Secondary antibody peroxidase conjugated AffiniPure Goat anti-rabbit (cat. #111-035-003, diluted 1:10,000) and anti-Guinea pig IgG (H + L) (cat. #106-035-003, diluted 1:10,000) from Jackson ImmunoResearch Laboratories, Inc. were used. SuperSignal Western Femto Sensitivity kit (Thermo Scientific, cat. # 34095) was used to develop the signal according to the manufacturer's instructions. The signal was captured by a Bio-Rad ChemiDoc MP Imaging System.

For CIII in-gel activity the purified protein samples were subjected to Colorless Native-PAGE (CN-PAGE) according to an established method[49]. Samples were run in pre-made 3-12% acrylamide gels in Bis-Tris (Invitrogen, cat. # BN2011BX10). 1x running buffer was used as cathode and anode buffer (Invitrogen, # BN2001). Cathode buffer without G-250 was used (Invitrogen, # BN2002). After loading the samples, starting voltage was 150 V for 15 minutes, then the voltage was increased to 250 V till the end of the run. Sample loading and run was done at 4 °C. The gel was incubated in the presence of 0.05% (w/v) diaminobenzidine (DAB) (Thermo Scientific, cat. # 34001) in 50 mM potassium phosphate buffer (pH 7.2) for several hours at room temperature[49]. For CIV activity the purified protein samples were subjected to BN-PAGE. Gels followed by incubation in the presence of horse heart Cyt *c* (Sigma, cat. # C2506-250MG) (0.05%, w/v) and 0.05% 3,3'-Diaminobenzidine (w/v) (Thermo Scientific, cat. # 34001) in 50 mM phosphate buffer (pH 7.2) according to an established method[49].

## Grids preparation, cryo-EM data collection, and image processing

Samples of purified WT SC were diluted 1:9 in 15 mM Tris-HCl, pH 7.2, and 20 mM KCl buffer to reduce digitonin and sucrose concentration and applied to 400-mesh copper grids containing ultrathin carbon support on lacey (Quantifoil, cat. # Q11459) followed by glow discharge at 30 mAmp for 15 sec. 5 μl samples were incubated for 1 min at 23 °C (100% humidity) inside a Vitrobot chamber, blotted for 3 sec and rapidly frozen in liquid ethane using a Vitrobot. Grids were stored at liquid nitrogen temperature for further analysis using cryo-EM. Data were collected with a Titan Krios microscope (Thermo Fisher Scientific) equipped with EPU 2.10.0.5 software operated at 300 kV (Supplementary Table 1 and Supplementary Fig. 4). A post-GIF K2 Summit direct electron detector (Gatan) operating in counting mode was used at a nominal magnification of X130,000 (pixel size of 1.07 Å) for image collection, and an energy slit with a width of 20 eV was used during data collection. A total dose of 49 eV/$Å^2$ fractionated over 35 frames was employed. Nominal defocus range set from −1.5 μm to −3.5 μm. Here, 20,253 micrograph movies were collected during multiple imaging sessions under identical conditions.

The WT tetrameric SC sample was processed with CryoSPARC (v3.2)[50]. Patch motion was used for frame alignment and exposure weighting with default parameters. The blob picker was initially used on a subset of images (300) to select a subset of particles, which were then used to generate a low-resolution template. Template particle picking was then performed, and the resulting particles were extracted using a 320 $Å^2$ box size. A total of 1,510,025 particles were selected. Multiple rounds of 2D classification followed, narrowing the dataset to 834,191 particles and then to 413,626 particles. An initial refinement was completed using C1 symmetry, resulting in a 4.14 Å density map. A non-uniform refinement followed by a local refinement with a tight mask further improved the resolution to 3.4 Å. Finally, a large heterogeneous 3D classification refinement routine was completed using four classes to sort the larger dataset (834,191 particles). These four initial models (density maps) contained the 3.4 Å structure, a density map that was soft-masked to only contain CIII, another density map that was soft masked to contain CIII and a noisy CIV, and a final map that appeared to only be noise. In total, 493,055 particles fell into the 3.4 Å map class. A final refinement with a dynamic mask was performed resulting in a 3.2 Å final density map.

Samples of the purified trimeric SC (III$_2$IV$_1$) isolated from the *CRD1Δ* strain were processed for microscopy, and data was collected (Supplementary Table 1 and Supplementary Fig. 13) as described for the tetrameric SC. A total of 12,006 micrograph movies were collected. The CRD1Δ dataset was imported with CryoSPARC and evaluated manually, at which point the number of micrographs was trimmed down to 9942 micrograph movies. Patch motion was used for frame alignment and exposure weighting with default parameters. Blob picker was initially used to select particles on a subset of images (300), which were then used to generate a low-resolution template. Template particle picking was then performed, and particles were extracted using a 320 $Å^2$ box size. An initial 1,926,302 particles were selected by the template picker, and after multiple rounds of 2D classification, the data were trimmed down to 638,401 particles. A strong presence of top and end-on views was visualized in the 2D class averages, as was smearing of density in the 3D reconstructions, indicating preferred orientation. When assessing particle orientations there were two distinct peaks in the data limiting the resolution and resolvability in the maps. Due to preferred orientation, an additional 4452 images were collected with a 30° tilt. This data was collected on a Titan Krios under similar microscope conditions, aside from the tilt. The tilt data was again processed with CryoSPARC in a similar fashion to the non-tilt data. Blob picker was used to select particles and after inspection and extraction, 545,397 particles were obtained. Two rounds of 2D classification followed, resulting in 186,312 particles being selected. The tilt

data was then added to the previous dataset to be processed in combination. A final heterogeneous refinement was performed on all the particles (tilt + non-tilt data) similar to the WT. The class that had CIII and CIV best resolved had 745,670 particles. This dataset was further refined using a tight mask. The resulting map was refined to 3.3 Å, improving upon the previous 3.5 Å map which did not include tilt data. Resolvability also improved, specifically in CIV.

## Model construction

Construction of the tetrameric SC model (III$_2$IV$_2$) began by fitting the previously reported structure of the III$_2$IV$_2$ mitochondrial respiratory supercomplex from *S. cerevisiae* (6HU9)[10]. This model, which included all proteins of interest and some ligands, was fitted into the tetrameric SC density map using the "Fit in Map" option in UCSF Chimera (v1.16)[51]. Qcr9 needed an additional adjustment (rigid body rotation), which was made with Chimera. Geometry minimization was run on the model to idealize the structure before refinement. Phenix (v 1.19.1-4122) real-space refinement followed[52] with ligand and Elbow restraints added and non-crystallographic symmetry (NCS) constraints turned off (due to non-strict C2 symmetry) and metal linking turned on. To optimize the CIII and CIV interface, subunit Cor1 of CIII and subunit Cox5a of CIV were submitted to GalaxyRefineComplex[53], a protein-protein complex refinement tool driven by interface repacking, via Galaxy Web. The top 10 docks all contained similar interface statistics and each of these models was fit to the density map and compared to the refined model. Docking 6 was chosen as the top model as it had the best fit-to-density and no clashes with other CIII and CIV subunits. This was then combined with the Phenix refined model.

Before iterating between real-space refinements and manual adjustments, PDB chains were renamed to account for pseudosymmetry (see Supplementary Table 4). As noted during the map reconstruction process, one CIV had less resolved density than the other, and thus the iterative model refinement process was focused on the stronger CIV density side (denoted as the reference-side). Throughout the modeling process, a trimer model consisting of the CIII (III$_2$) dimer and one CIV on the reference-side was refined. Once refined, the opposite CIV was assessed and modeled. Additional density was present in various regions of the map, with a majority in CIII. To isolate this additional density, a difference map was created from the model. This difference map revealed a noisy detergent band around the transmembrane portion of the supercomplex, as well as discrete densities within CIII and at the interface of CIII and CIV. To model these densities, previously resolved SCs and CIII (6YMX, 1KB9, 3CX5, 6Q9E, 6GIQ[11,17,18,44,45]) were fit into the density map using Chimera. This excess density was modeled with four UQ6 ligands, CN5 (CL) at the 2-fold of CIII, and CL in similar positions to 6YMX. Moreover, excess density was found at the location where PE was modeled in 6YMX. This was renamed to PEF, rigid-body fit with Chimera, and manually adjusted in COOT (v0.9.5)[54]. Ligand restraints (.cif files) were then generated in Phenix[43] for ligands that were not defined in the Phenix ligand library. Furthermore, Elbow restraints were computed using phenix.metal coordination, ensuring that bonds with metal ions were maintained during model refinement. Finally, multiple iterations of Phenix real-space refinement (using the above restraints and NCS symmetry turned off) and COOT adjustments were done to improve the fit-to-density and model geometry of the trimer complex. Throughout the refinements, ligands were manually altered to better fit the weaker density. Once complete, the refined reference CIV model was duplicated and rigid-body-fit into the opposite-side CIV density. An additional iteration of real-space refinement, without the aid of NCS restraints, was run to improve fit-to-density and obtain B-factor values for the tetramer before model interpretation (Supplementary Fig. 17).

Modeling of the CRD1Δ density map began by independently fitting the optimized WT protein subunits from CIII and CIV into the

density map with Chimera. Iterative refinement of the protein complex was performed using Phenix real-space refinement and COOT. Once complete, the CRD1Δ SC map was masked using the protein model to reveal unmodelled density. Hemes, appropriate ligands (PEF, PCF) and cofactors were rigid body fit to the unmodeled parts of the CRD1Δ SC density map. When analyzing the masked density and the corresponding model, it was noted that the CDL locations did not have one-to-one matching density. The density at these positions revealed two tails with one strong headgroup. Anionic phospholipid PGT, was then fit into the CDL positions. Phenix was then used to generate ligand and metal bond restraints, and real-space refinement followed. As with the WT, COOT and Phenix were iteratively used to optimize the modeled complex. Ligand density that was disordered, specifically tail density was truncated. For the chain/subunit names and ligands see Supplementary Table 6.

To further analyze the PGT fitting, PEF, PCF, and CDL were all fit into the PGT density locations using Chimera. One iteration of Phenix real-space refinement was completed with the ligands in the isolated PGT density. A calculated density map was computed for each of the ligand groups and the threshold was set to be equivalent for each ligand group. The cross-correlation was computed for the six locations between the experimental masked density and the calculated density (Supplementary Table 7)[55,56] with PGT density showing the highest correlation.

For both the WT and CRD1Δ structures, visualization and analysis of the final models were done with UCSF Chimera, COOT, Phenix, and Molprobity[57]. Moreover, LigPlot+ (v2.2)[58] was used to analyze the ligands at the CIV-CIII interface. Settings in LigPlot+ were changed such that hydrogen bond D-A pairs had a maximum distance of 4 Å and non-bonded interacting residues were only between hydrophobic-hydrophobic contacts.

## Reporting summary

Further information on research design is available in the Nature Portfolio Reporting Summary linked to this article.

## Data availability

The data that support this study are available from the corresponding authors upon request. The cryo-EM maps have been deposited in the Electron Microscopy Data Bank (EMDB) under accession codes EMD-27940 (WT) and EMD-28011 (CRD1Δ). The atomic coordinates have been deposited in the Protein Data Bank (PDB) under accession codes 8E7S (WT) and 8EC0 (CRD1Δ). The following public data bases were used for the following structures: 6HU9 [https://doi.org/10.2210/pdb6HU9/pdb], 6YMX [https://doi.org/10.2210/pdb6ymx/pdb], 1KB9 [https://doi.org/10.2210/pdb1kb9/pdb], 3CX5 [https://doi.org/10.2210/pdb3cx5/pdb], 6Q9E [https://doi.org/10.2210/pdb6Q9E/pdb], 6GIQ [https://doi.org/10.2210/pdb6Q9E/pdb]. Source data are available as a Source Data file. Source data are provided with this paper.

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

## Acknowledgements

This work was supported in part by NIH/NIGMS (R01GM115969) and the John S. Dunn Research Foundation to W.D., NIH/NIGMS (P01GM063210) to M.L.B, and NIH/NIGMS (R01GM072804) to I.I.S. Cryo-EM experiments were performed using resources of the Cryo-EM Core Facility at UTHealth. Titan Krios G3 microscope (Thermo Fisher Scientific Inc.) subsidized by CPRIT Core Facility Award RP190602 was used for cryo-EM data acquisition. We also thank Dr. Rebecca Berdeaux and Dr. Antonio Soares in the Department of Integrative Biology and Pharmacology at McGovern Medical School for use of their Oxygraph-2k (supported by NIH NIDDK (R01DK092590) and American Diabetes Society (1-18-IBS-050) to R.B.) and assistance in running oxygen consumption assays.

## Author contributions

W.D and E.M. conceptualization and supervised the project and designed the research. V.K.P.S.M. and E.I.B. did sample preparation and biochemical analysis. V.K.P.S.M. and S.A. did cryo-EM data collection. G.F., I.I.S., and S.A. did preliminary data processing. C.F.H. and M.L.B. did the final data processing and constructed the models. E.M., C.F.H., M.L.B., and W.D. did structure analysis and interpretation. G.C.S. did a

mass spectral analysis. E.M., W.D., C.F.H., and M.L.B. wrote the manuscript. I.I.S. provided important insight into the writing of the manuscript. All authors provided information and/or critical comments during manuscript preparation. W.D. (william.dowhan@uth.tmc.edu), E.M. (eugenia.mileykovskaya@uth.tmc.edu), and M.L.B. (matthew.l.baker@uth.tmc.edu) are co-corresponding authors.

## Competing interests

The authors declare no competing interests.
