## [Peer Review File · Nature Communications]

Structural insights into cardiolipin replacement by phosphatidylglycerol in a cardiolipin-lacking yeast respiratory supercomplex using cryo-EMReviewers' Comments:

Reviewer #1:

Remarks to the Author:

The manuscript by Hryc et al presents the cryoEM structures of the yeast III-IV supercomplex isolated from a wild-type and a cardiolipin-lacking mutant strain to test the importance of lipids, and in particular cardiolipin, in stabilizing the SC.

The structure derived from the wild-type strain is overall similar to those already published, with data at high resolution on the CIII dimer core but lower resolution towards the periphery of the SC and CIV. Cardiolipin is seen to mediate interactions between CIII and CIV in the membrane domain and the model confirms the remarkable conservation of the cardiolipin sites as reported by at least three other research groups. However, of importance is the observation of quinone at the Qi site of CIII (as well as at Qo) which most likely arise from the method used to isolate the SC.

In absence of cardiolipin, the architecture of the SC remains the same and phosphatidylglycerol, the direct precursor of cardiolipin, could be modeled in similar positions, filling the space between CIII and CIV within the membrane domain and stabilizing the SC with slight differences in the CIII and CIV residues involved.

The manuscript is well written but appears slightly rushed or unfinished for aspects that are detailed below. My main criticism is the lack of robust evidence (in the form of gels and activity assays) to back up the claim of a difference of stability or behavior of the SC that would be solely due to the presence or absence of cardiolipin vs PG. There is too much reliance on past (and sometimes very old) publications which is not justified in the case of yeast where there is no limitation on amount of material available. Knowledge in the field has advanced greatly and there is clear recognition that details in purification protocols from a lab to another (or even experimentalist within the same lab) can make all the difference. In addition, it is unfortunate that the two SCs have been isolated with completely different protocols (WT by sucrose gradient, CL-lacking mutant by metal-affinity) resulting in significant differences to the SCs composition (loss of the Qcr10 subunits on CIII and no quinone seen at Qi for the mutant) in addition to presence or absence of CL. In this context, a thorough characterization of the solubilised mitochondria prior to any SC purification is indispensable.

Major points:

p1. Abstract. There is no data in the manuscript that demonstrate any 'reduced stability' of the yeast supercomplex in the absence of cardiolipin or 'destabilization of supercomplex formation by phosphatidylglycerol'. Instead it could be seen as rather remarkable that the same SC can form with PG! The analogy between the present study and potential effect of cardiolipin loss in Barth Syndrome patients is also far-fetched in that case as the yeast III-IV SC is completely different in architecture (and most likely biogenesis) to the mammalian one and extension of the effect of PG on other SCs remains to be investigated.

p2, l-64 to 70, including 'Digitonin extracts of wild type (WT) yeast mitochondria when displayed by Blue Native (BN)-PAGE reveals almost exclusively a tetrameric SC (III₂IV₂)'. This section is not correct. Not only does the organization of respiratory complexes in the membrane vary depending on growth conditions (and other factors), the ratio of 'tetramers' vs 'trimers' (and free CIII and CIV) directly correlates with the ratio of CIII and CIV in the membrane. The newly added gels (new SI figure 12) are from the purified SCs and so do not represent the organization of the total CIII and CIV in the membrane. They only show what is in the sample used for structure determination! If CIII and CIV cannot be accurately quantitated by other means, a minimum requirement would be to add BN-PAGE (and/or Coomassie stained CN-PAGE) of the solubilised membrane prior to SC purification. Only then could one fully appreciate the effect of the mutant on CIII and CIV organization. This is of particular importance as the behavior of those proteins in the CL-lacking strain is different to in ref 16 (see below).

p3, l-89. 'The SC obtained by this method is active and demonstrates a low level of the ubiquinol oxidase activity without any addition of Cyt c.' There is no new data associated to this statement so it seems to solely refer to ref 9. A new activity assay should be performed to (also) give full activity in the presence of added Cyt c (absent from ref 9) instead of 'low level of' so the data can be compared to other structural studies and, more importantly, to the activity of the SC isolated from the CL-lacking strain.

p7, l-230 'The lack of significant formation of the tetrameric SC in the absence of CL' and 'The apparent weaker interactions of PG with the individual complexes at the interface appears to be the basis for lack of tetrameric SC formation'. Again, information is missing to support this claim. Especially as in ref 16, there is no difference in SC migration pattern on native gels ran from solubilised mitochondria from WT and the CL-lacking mutant. 'Tetramers' clearly form in ref 16 in the CL-lacking strain and with same ratio of 'tetramers' vs 'trimers'. Again, the stoichiometry of the SC is known to directly correlate with the ratio of CIII and CIV in the membranes. Are CIII and CIV expressed at the same level in both strains? A BN-PAGE (and/or Coomassie stained CN-PAGE) of the solubilised mitochondria, in addition to the new SI fig 12 showing purified SC samples, is required to evaluate 'tetramer' and 'trimer' and free CIV and free CIII in both WT and CL-lacking mutant.

p7, l-234. 'the basis for lack of tetrameric SC formation under more stringent conditions of BN-PAGE.' Couldn't the stability (or loss of) the SC in the CL-lacking mutant vs WT be tested by running parallel sample lanes of the purified SCs (or even simpler, solubilised mitochondria) after incremental addition of another detergent known to disrupt SCs? The PG stabilized SC should be seen to dissociate at lower additive concentration compared to CLs stabilised SC. That should show on a BN-PAGE (or a Coomassie stained CN-PAGE, as preferred).

There is no image processing pipeline in SI and it is difficult to assess if the quality of the maps could have been further improved. In fact, the method is incomprehensible in places (p14, l-447 'Throughout the modeling process, a trimer model consisting of the CIII (III2) dimer and one CIV on the reference-side was refined.' What does that mean? Did you find classes for the 'trimer' or was the density for one CIV subtracted from the particles? Explain).

Other points:

The title could better encapsulate the novelty of the study about PG compensation in absence of cardiolipin.

p2, l-39. It is unclear what is meant by 'However, these are stable structures, which do not undergo dissociation/association of the subunits from the larger complexes.' That binding of/to CL is irreversible? Please clarify.

p2, l-50. 'CL molecules at the interface between CIII and CIV in yeast (10,15,16) and mammalian systems (17-19).' Where in mammalian SCs are the interface CLs? Between each individual complexes or only between some? It's unclear if the sentence refers to between CIII and CIV only (in which case ref 18 is not well chosen) or between all individual complexes in general. Please clarify. Most importantly, there is no reference to the recently published structures of the mammalian III-IV SCs with SCAF1. In addition, the EM map in ref 17 was at 19A resolution and so cannot provide any information as to where the CLs are! They identified the presence of CL in the SC from lipid extracts but didn't 'verify CL molecules at the interface' (those CL could have been anywhere).

p3, l-70. 'which has minimized the importance of a CL or lipid requirement in SC formation in favor of protein-protein interactions being the primary basis for SC formation (1).' This isn't correct and clearly an outdated statement (ref 1 is from 2009!). The importance of lipids and the special role of CL in SC formation/stabilization has been widely recognized and in particular in yeast where CLs mediate all

IMM interactions between the two complexes.

p3, l-75. 'and whole respiratory chain kinetic properties are altered (13).' This statement should be developed to explain what the data in ref (13) actually show in light of current status of knowledge.

p3, l-96. Description of overall protein subunit composition for CIII and CIV should be included.

p3, l-97. 'The reconstruction also revealed that one CIV was better resolved.' Why is that? Please, provide an explanation.

Figure 1. All panels. It is difficult to understand what the authors want to show with the 'grey area' which is confusing and should be removed as there is no possible comparison between the two datasets.

p4, l-112 to 114. Add in the text the label used for each quinone as per the model (603J l-112? and 606J l-114?) for clarity.

p4, l-129. 'Further studies of the SC purified by sucrose gradient centrifugation and its red-ox state are required to provide a detailed explanation of this arrangement.' It is implied here that there may be an effect of the SC purification method used (sucrose gradient vs metal-affinity) and indeed there may well be, as subsequently demonstrated with the purification of the CL-lacking mutant. This sentence could be moved to the discussion where the effect of differences in purification methods used to purify the WT and mutant strain should be discussed. Delipidation of membrane proteins on metal-affinity purification step is well documented and it should be noted that the CL-lacking SCs could be stabilized by other lipids lost with the harsher purification method or when solubilised from the membrane.

p5, l-138. Typo in 'localizarion' and 'structue'.

p5 in 'Localizarion of Cardiolipin in the WT SC structue'. Are all the CL described here also in the previously resolved cryoEM SC structures? Please, clarify.

p8, l-248. 'Analysis of extracts from *crd1Δ* mitochondria under more stringent conditions shows high levels of free CIII and CIV'. I cannot seem to find a gel in Ref 12 in support of that statement.

p8, l-255. 'resulting in increased free individual complexes.' Again, ref 16 doesn't support more free CIII and CIV in the absence of CL.

p12, l-379. 'Grids for cryo-EM were prepared as previously described (9).' Is this only true for the WT? Specify.

p14, l-453. 'previously resolved yeast SCs and CIII (6YMX, 1KB9, 3CX5, 6Q9E, 6GIQ 11,16,18,41,42)'. Ref 18 is not from yeast.

There is no mention of data deposition (EM maps and models) in appropriate data banks.

SI Table 2. What is being referred to as 'chains' in the table? 64 chains for the tetramer and 48 for the trimer? Please explain and clarify.

Reviewer #2:

Remarks to the Author:

The manuscript by Hryc et al describes the structure of the intact CIII₂/CIV₂ supercomplex from the

yeast respiratory chain as compared to a CIII2/CIV1 complex isolated from a cardiolipin-deficient yeast mutant. The structure of the CIII2/CIV2 supercomplex revealed ubiquinone-6 molecules bound all four quinone binding sites and a series of cardiolipin molecules, in particular at the CIII/CIV interface. In the CIII2/CIV1 complex, fewer quinones were bound and the cardiolipin molecules were replaced by phosphatidylglycerol, which showed less interaction with the CIII and the CIV part of the complex.

The authors discuss their findings in relation to the role of cardiolipin for supercomplex stability and medical conditions involving impaired decrease cardiolipin levels and/or impaired respiratory supercomplex function.

These findings are interesting and will promote the field.

The authors should address the following comments:

1.The authors mention that the purified CIII2/CIV2 used for structure determination was enzymatically active, but refer to one of their previous papers. Please state activity values in the current manuscript for both CIII2/CIV2 and CIII2/CIV1.

2.The authors argue that the stability of supercomplexes such as CIII2/CIV2 is reduced in absence of cardiolipin (e.g.: lines 68-72 and elsewhere in the manuscript).

Did the authors experimentally confirm decreased stability for the purified CIII2/CIV1 complex, e.g.: by isothermal calorimetry, activity assays, etc? Does CIII2/CIV1 disintegrate into CIII and CIV over time (faster than CIII2/CIV2)?

3.The phosphatidylglycerol content of the cardiolipin-deficient yeast mutant was determined by mass spectrometry (lines 165-167). Please show the mass spectrometry result in an appropriate supplementary display item.

4.The Qcr10 subunit apparently was lost during purification of CIII2/CIV1, but not of CIII2/CIV2. For the non-specialist reader it is difficult to assess which impact the presence or absence of this subunit may have on supercomplex stability. Please clearly indicate the location of Qcr10 in the CIII2/CIV2 structure.

Textual:

Line 53: "energy production", please rephrase

Line 123: "the"

Line 129: "red-ox"

Line 207: "this"

RESPONSE TO REVIEWER COMMENTS

We thank the reviewers for their comments and suggestions, which allowed us to strengthen the manuscript and focus on the primary message, *i.e.* the role of anionic phospholipids in supporting respiratory SC formation as now reflected in the new title. Our previous emphasis on stability/formation of the SC detracted from the main message relating to the importance of anionic phospholipids. Whether lack of CL results in lower stability or lack of formation of the tetrameric SC is a question that still requires investigation but is not the subject of this manuscript. There have been numerous reports detailing a change in distribution of tetrameric and trimeric SC forms as well as the level of individual respiratory complexes from WT and Δ CRD1 cells since our original report (Ref 13). Therefore, we provide new experiments that demonstrate the high presence of the trimeric SC and individual complexes in mitochondrial extracts of the Δ CRD1 strain as compared to the almost exclusive presence the tetrameric SC from the WT strain. In fact, under our conditions, the tetramer is almost completely absent in the former strain. Again, whether this is due to stability or lack of formation is not known. Our results as well as that of others (Ref 11) indicate that one of the two CIV's in the WT structure differs in resolution, which may be due to differences in flexibility and therefore compromise tetramer formation in the absence of CL. Since our focus is on the role of anionic lipids in SC formation rather than a detailed analysis of lipid dependent activity, we did not focus on the latter as explained below. The points we hope are now evident are as follows. 1) An anionic phospholipid (CL or PG) is required for SC (trimer or tetramer) formation. 2) There are significance differences between interaction of CL and PG with the CIII and CIV, which may account for the differences in SC formation and/or stability. 3) Both anionic lipids interact with positive domains at the CIII/CIV interface thereby dampening charge repulsion between individual complexes. 4) Anionic phospholipid binding provides nucleation sites for additional phospholipids to form a lipid domain, which further stabilizes SC formation. 5) A combination of protein-protein and lipid protein-interaction supports the formation SCs.

Reviewer #1 (Remarks to the Author):

The manuscript by Hryc et al presents the cryoEM structures of the yeast III-IV supercomplex isolated from a wild-type and a cardiolipin-lacking mutant strain to test the importance of lipids, and in particular cardiolipin, in stabilizing the SC. The structure derived from the wild-type strain is overall like those already published, with data at high resolution on the CIII dimer core but lower resolution towards the periphery of the SC and CIV. Cardiolipin is seen to mediate interactions between CIII and CIV in the membrane domain and the model confirms the remarkable conservation of the cardiolipin sites as reported by at least three other research groups. However, of importance is the observation of quinone at the Qi site of CIII (as well as at Qo) which most likely arise from the method used to isolate the SC.

Our important observation is the ubiquinone occupies the Qo sites while previously published structures only showed ubiquinone at Qi sites. This is a side observation not related to the focus of the paper. However, it is an important observation.

In absence of cardiolipin, the architecture of the SC remains the same and phosphatidylglycerol, the direct precursor of cardiolipin, could be modeled in similar positions, filling the space between CIII and CIV within the membrane domain and stabilizing the SC with slight differences in the CIII and CIV residues involved.

The manuscript is well written but appears slightly rushed or unfinished for aspects that are detailed below. My main criticism is the lack of robust evidence (in the form of gels and activity assays) to back up the claim of a difference of stability or behavior of the SC that would be solely due to the presence or absence of cardiolipin vs PG. There is too much reliance on past (and sometimes very old) publications which is not justified in the case of yeast where there is no limitation on amount of material available. Knowledge in the field has advanced greatly and there is clear recognition that details in purification protocols from a lab to another (or even experimentalist within the same lab) can make all the difference. In addition, it is unfortunate that the two SCs have been isolated with completely different protocols (WT by sucrose gradient, CL-lacking mutant by metal-affinity) resulting in significant differences to the SCs composition (loss of the Qcr10 subunits on CIII and no quinone seen at Qi for the mutant) in addition to presence or absence of CL. In this context, a thorough characterization of the solubilised mitochondria prior to any SC purification is indispensable.

Initially, we also believed that the difference in purification is a negative point, but when we obtained our structures and models and discovered that these two SCs purified by two different methods have a nearly identical organization, *i.e.* similar mutual orientation of CIII and CIV, similar organization of subunits and most important the extra-density in the mutant in positions similar to extra-densities for CL in WT SC, we concluded that the difference in the purification makes our observation/discovery for PG replacing CL in the mutant structure even a stronger discovery. As noted in the revised manuscript under the new section dealing with purification (lines 103-113), we reviewed previous reports using various means of purification that yielded identical structures for all WT SC.

As to Qcr10p subunit, it is not at the interface of CIII and CIV, and there are no significant connections with CL molecules in the WT. Its absence did not influence the position of other subunits in the mutant when compared to the WT structure. Lack of strong interaction of Qcr10 with CIII is a well-established fact. The crystal structure of the yeast CIII obtained by Hunte et al. (Ref 41) does not contain Qcr10. Since this subunit does not have strong interaction with CIII and was partially lost during purification of CIII for crystallization, it was completely deleted to obtain crystals (lines 131-134). The organization of the protein components of the WT and Δ CRD1 SCs and nearly identical except for the absence of Qcr10 in the latter. Therefore, we maintain that the only differences are in ligand association, which is the focus of this manuscript.

Major points:

p1. Abstract. There is no data in the manuscript that demonstrate any 'reduced stability' of the yeast supercomplex in the absence of cardiolipin or 'destabilization of supercomplex formation by phosphatidylglycerol'. Instead, it could be seen as rather remarkable that the same SC can form with PG! The analogy between the present study and potential effect of cardiolipin loss in Barth Syndrome patients is also far-fetched in that case as the yeast III-IV SC is completely different in architecture (and most likely biogenesis) to the mammalian one and extension of the effect of PG on other SCs remains to be investigated.

As noted above, the issue of formation/stability has been mostly removed and still requires further investigation but is not the focus of the revised manuscript. We acknowledge the differences between the yeast and mammalian structures. However, the role of anionic

phospholipids in supporting SC formation should be very similar. As to Barth Syndrome, we have removed this point from the manuscript.

p2, l-64 to 70, including 'Digitonin extracts of wild type (WT) yeast mitochondria when displayed by Blue Native (BN)-PAGE reveals almost exclusively a tetrameric SC (III₂IV₂)'. This section is not correct. Not only does the organization of respiratory complexes in the membrane vary depending on growth conditions (and other factors), the ratio of 'tetramers' vs 'trimers' (and free CIII and CIV) directly correlates with the ratio of CIII and CIV in the membrane. The newly added gels (new SI figure 12) are from the purified SCs and so do not represent the organization of the total CIII and CIV in the membrane. They only show what is in the sample used for structure determination! If CIII and CIV cannot be accurately quantitated by other means, a minimum requirement would be to add BN-PAGE (and/or Coomassie stained CN-PAGE) of the solubilised membrane prior to SC purification. Only then could one fully appreciate the effect of the mutant on CIII and CIV organization. This is of particular importance as the behavior of those proteins in the CL-lacking strain is different to in ref 16 (see below).

Our statement concerning almost exclusively tetramer in the WT extracts is supported by new BN-PAGEs (Supplementary Fig. 1). Line 115-120. This is like our previous reports in Ref 9, 13, 14 as well as (Ref 39 and Schagger and Pfeifer (2000) Embo J 19, 1777). Yes, there is variability based on growth conditions, but this is due to the ratio of CIII to CIV which comes into play in glucose grown cells where the level of CIV is suppressed versus CIII (Schagger above) resulting in accumulation of mainly free CIII. Under all growth conditions WT mitochondrial extracts in the above papers and our new results show only trace amounts of the trimer and free complexes. In the mutant there is accumulation of free CIII and CIV, which based on the WT case should not occur if there is no problem with assembly or stability, i.e. free individual complexes should associate with only the excess complex remaining free. The reviewer refers to Ref 17 (previous Ref 16) as newer data that contradicts our statement concerning the mutant extracts contain primarily trimer and free complexes. This is the only report showing high levels of the tetramer in the mutant extracts for which we have no explanation and rely on our old and new data. However, even in the case of Ref 17 (Fig. 2), there is an increase in the trimer as well as free CIII and CIV over that seen in the WT as detected by antibody probes.

We provide new Western blots probed with antibodies against for CIII and CIV of the digitonin extracts from the WT and Δ CRD1 strains prior to SC purification. These demonstrate that in contrast to the WT extract, which shows mostly the SC tetramer, the extract from the mutant mitochondria contains mostly trimer and high amounts of free CIII and CIV (Supplementary Fig. 1). It should be noted that the amount of blue dye used in BN-PAGE also affects the distribution of complexes from the Δ CRD1 mitochondria. In our original papers (Ref 13 and 14) we used higher levels of blue dye than in the current data resulting in only individual respiratory complexes.

p3, l-89. 'The SC obtained by this method is active and demonstrates a low level of the ubiquinol oxidase activity without any addition of Cyt c.' There is no new data associated to this statement so it seems to solely refer to ref 9. A new activity assay should be performed to (also) give full activity in the presence of added Cyt c (absent from ref 9) instead of 'low level of' so the data can be compared to other structural studies and, more importantly, to the activity of the SC isolated from the CL-lacking strain.

See Lines 114-138).

The focus of the paper is lipid-dependent formation of SCs so that a detailed analysis of activity is not of central importance as it was in Moe et al. (Ref. 21). Therefore, as in Rathore et al. (Ref 11, in Nat Struct Bio where the focus was the structure of the SC with no activity data reported), we initially provided minimal activity data (qualitative in-gel assays) and relied on previous results published by us (Ref 9 and 14). Just because the results are 10 years old does not minimize the reliability of the data. We note the previous results showing oxygen consumption being the same for WT and Δ CRD1 mitochondria (Ref 14) and the oxygen consumption for the WT purified SC as 1.0 μ mole O₂/min/mg SC; the result is in Ref 9 (see p. 23099, first 3 lines in left column). WT SC in Ref 9 was purified by the same method as in the current manuscript. Additionally, we calculated the activity of the WT SC isolated in Ref 21 (using Flag-tagged CIV) from the oxygen consumption curve in Fig. S3, which was 0.9 μ mole O₂/min/mg SC as compared to our previously determined value of 1.0 μ mol O₂/min/mg. The authors stated that their value was the same as reported in Ref 17 (isolated by Flag-tagged CIII).

However, to address the reviewers concerns we have measured activities (see below) as best we could given the limited amount of remaining sample used to determine the structures. Much of the delay in responding is due to our search for an instrument to measure oxygen consumption. We no longer have a Clark-type electrode but were able to locate a Oxygraph-2k high resolution respirometry system, which has been used for cells and organelles, but there are no reports of its use for purified enzyme samples. Significant time was devoted to finding an instrument and adapting its use for our purpose.

We present BN-PAGE figures accompanied with Western blots of the extracts (Supplementary Fig. 1). We now present oxygen consumption results for isolated mitochondria (Supplementary Fig. 2), which are in reasonable agreement with our previous results (different genetic backgrounds and different instrument for assays). A comparison of coupled SC activity (reduced UQ and Cyt c) between the WT and mutant mitochondrial extracts provides no information on comparable activities between the tetrameric and the trimeric SCs in the extract. Since the WT extract is almost exclusively tetramer, such an assay would measure coupled activity. However, the mutant extract contains a complex mixture of some tetramer and mostly the trimer and large amounts of free CIII and CIV. Activity is a composite of coupled electron transport within the SCs and electron transport between freely diffusing complexes.

We made measurements on the limited amounts of SCs available (3 for WT and only one for Δ CRD1, see also Methods (Lines 360-368) and Supplementary Fig. 2.) The WT SC measurements were in good agreement with our previous results. However, the CRD1 Δ SC single measurement (no previous report) was much lower. This is consistent with the lower consumption by CRD1 Δ mitochondria and the low in-gel activity of the CRD1 Δ CIII compared to the WT CIII (Supplementary Fig. 3) as we reported previously. This is most likely due to the known 40% reduction (Ref 42) in CIII activity when lacking Qcr10 coupled with initial lower activity of the Δ CRD1 mitochondria. Whether substitution of PG for CL further reduces activity is beyond the scope of our focus, which would require making mutants lacking Qcr10 in the WT strain followed by purification and comparison with activity of the SC purified from the mutant strain.

We acknowledge that final activity measurements might be of interest to readers. However, further studies comparing the activities of the purified SCs would not change the important conclusions we provide on lipid-dependent organization of SCs.

p7, I-230 'The lack of significant formation of the tetrameric SC in the absence of CL' and 'The apparent weaker interactions of PG with the individual complexes at the interface appears to be the basis for lack of tetrameric SC formation'. Again, information is missing to support this claim. Especially as in ref 16, there is no difference in SC migration pattern on native gels ran from solubilised mitochondria from WT and the CL-lacking mutant. 'Tetramers' clearly form in ref 16 in the CL-lacking strain and with same ratio of 'tetramers' vs 'trimers'. Again, the stoichiometry of the SC is known to directly correlate with the ratio of CIII and CIV in the membranes. Are CIII and CIV expressed at the same level in both strains? A BN-PAGE (and/or Coomassie stained CN-PAGE) of the solubilised mitochondria, in addition to the new SI fig 12 showing purified SC samples, is required to evaluate 'tetramer' and 'trimer' and free CIV and free CIII in both WT and CL-lacking mutant.

See discussion above on this point. We provide new BN-PAGE of the digitonin extracts from the WT and mutant mitochondria (Supplementary Fig. 1). As to stoichiometry, the reviewer is correct in relation to the level of individual complexes affecting SC formation. However, if only one of the individual complexes is limiting then only the excess free respiratory complex should have been observed as observed in glucose growth cells (Schagger 2000 above). The important point is that there are large amounts of free CIII and CIV in our mutant extract gels as well as those in (Ref 17). If there were no problems with either stability or formation of the SC in the mutant, then there would be very low levels of both free respiratory complex or just the complex in excess but not both.

p7, I-234. 'the basis for lack of tetrameric SC formation under more stringent conditions of BN-PAGE.' Couldn't the stability (or loss of) the SC in the CL-lacking mutant vs WT be tested by running parallel sample lanes of the purified SCs (or even simpler, solubilised mitochondria) should show on a BN-PAGE (or a Coomassie stained CN-PAGE, as preferred). after incremental addition of another detergent known to disrupt SCs? The PG stabilized SC should be seen to dissociate at lower additive concentration compared to CLs stabilised SC. That

As noted in the introduction to our response, stability/formation (reported and studied by us and others) is not the focus of this manuscript. It is clear from the new BN-PAGEs that extracts from the mutant contain primarily trimer and individual complexes. The presence of the latter is consistent with the altered kinetics for whole chain activity for intact mitochondria we presented in (Ref 14), which indicates an increase in free complexes in the mutant mitochondria.

There is no image processing pipeline in SI and it is difficult to assess if the quality of the maps could have been further improved. In fact, the method is incomprehensible in places (p14, I-447 'Throughout the modeling process, a trimer model consisting of the CIII (III2) dimer and one CIV on the reference-side was refined.' What does that mean? Did you find classes for the 'trimer' or was the density for one CIV subtracted from the particles? Explain).

To address the lack of clarity in image processing, we have added image processing pipelines for both the WT (Supplementary Fig. 4) and mutant (Supplementary Fig. 13). As for the modeling process question, because one CIV was better resolved (as discussed in the Results section) we only focused on iteratively refining a timer model (Lines 140-150). The weaker resolved CIV (opposite-side) was not the primary focus in the modeling process until the end, where the reference CIV model was positioned in the opposite-side density and refined a single time with Phenix real-space refinement. This process ensured that the model was not overfit into the weaker CIV density.

Other

points:

The title could better encapsulate the novelty of the study about PG compensation in absence of cardiolipin.

Thank you for this suggestion. Title changed and emphasis on CL replacement of PG in the revised manuscript was very helpful.

p2, l-39. It is unclear what is meant by 'However, these are stable structures, which do not undergo dissociation/association of the subunits from the larger complexes.' That binding of/to CL is irreversible? Please clarify.

Removed from manuscript

p2, l-50. 'CL molecules at the interface between CIII and CIV in yeast (10,15,16) and mammalian systems (17-19).' Where in mammalian SCs are the interface CLs? Between each individual complexes or only between some? It's unclear if the sentence refers to between CIII and CIV only (in which case ref 18 is not well chosen) or between all individual complexes in general. Please clarify. Most importantly, there is no reference to the recently published structures of the mammalian III-IV SCs with SCAF1. In addition, the EM map in ref 17 was at 19A resolution and so cannot provide any information as to where the CLs are!

Ref 17 identified the presence of CL in the SC from lipid extracts but didn't 'verify CL molecules at the interface' (those CL could have been anywhere). In mammalian SC CL is between all individual complexes so the general statement is correct. Ref 17 is removed. Lines 51-52

p3, l-70. 'which has minimized the importance of a CL or lipid requirement in SC formation in favor of protein-protein interactions being the primary basis for SC formation (1).' This isn't correct and clearly an outdated statement (ref 1 is from 2009!). The importance of lipids and the special role of CL in SC formation/stabilization has been widely recognized and in particular in yeast where CLs mediate all IMM interactions between the two complexes. We need to explain our statement to rev.

Yes, in general CL is accepted as important in SC formation, but we now list previous ref 16 (now Ref 17) as casting doubt in favor of protein-protein interactions as primary (lines 74-82) as Rev 1 has also suggested.

p3, l-75. 'and whole respiratory chain kinetic properties are altered (13).' This statement should be developed to explain what the data in ref (13) actually show in light of current status of knowledge

We have expanded discussion in the text as requested. Lines 80-82

p3, l-96. Description of overall protein subunit composition for CIII and CIV should be included.

Tables provided (Supplementary Table 3 and 4) which outlined the various protein subunits that comprise the structure. All cofactors and ligands associated with each subunit are also indicated in these tables. Supplementary Fig. 8 shows the structure and position of each individual subunit of CIII and CIV in the SCs. This provides sufficient

information for understanding structural and functional property of each individual subunit of CIII and CIV in the SC and serves as the descriptive information.

p3, I-97. 'The reconstruction also revealed that one CIV was better resolved.' Why is that? Please, provide an explanation.

We thank the reviewers for this great question. This structural artifact has been seen in other SCs (Ref 11). Any explanation would be highly hypothetical. We could speculate. It appears that the trimer structure is more stable than the tetrameric structure since the former predominates in the mutant. It is possible that the association of one CIV has a subtle effect on the stability of the second CIV association.

Figure 1. All panels. It is difficult to understand what the authors want to show with the 'grey area' which is confusing and should be removed as there is no possible comparison between the two datasets.

We show the grey area to provide the reader with a better understanding of the experimental density map. By showing the low-threshold density, the reader has a better grasp of map regions which have less signal. While it could be removed, we would rather not, to help readers better visualize our density maps. An explanation of the possible reason for the added density is in the legend. Please note that the CRD1 Δ structure shows fewer low-density areas.

p4, I-112 to 114. Add in the text the label used for each quinone as per the model (603J I-112? and 606J I-114?) for clarity.

Done. Lines 160-164

p4, I-129. 'Further studies of the SC purified by sucrose gradient centrifugation and its red-ox state are required to provide a detailed explanation of this arrangement.' It is implied here that there may be an effect of the SC purification method used (sucrose gradient vs metal-affinity) and indeed there may well be, as subsequently demonstrated with the purification of the CL-lacking mutant. This sentence could be moved to the discussion where the effect of differences in purification methods used to purify the WT and mutant strain should be discussed. Delipidation of membrane proteins on metal-affinity purification step is well documented and it should be noted that the CL-lacking SCs could be stabilized by other lipids lost with the harsher purification method or when solubilized from the membrane.

Based on our data for CL and PG presence and positions, there is no effect on lipids. In the case of UQ6 the redox state is important. This is not about the difference in the UQ6 position between our WT structure and our CRD1 Δ structure. Position of UQ6 in the CRD1 Δ structure at 2 Qi binding sites is the same as published for the WT SC structures 6YMX and 6GIQ. This is about our WT, which was purified without any kind of chromatography. Our statement about future studies of the red-ox state is correct and sufficient for the paper. This is an additional interesting result for our future study not related to the role of anionic lipids in SC formation. The 6GIQ SC (Ref 11) was affinity purified with anti-FLAG resin (Qcr7-FLAG) and contained CL at the same sites we observed for our sucrose gradient purified SC. Therefore, a difference in UQ occupancy is unlikely due to delipidation during purification. The lack of UQ in the Qp site of the previously reported WT SCs is explained by the lower affinity of the oxidized UQ to Qp site and high affinity of

oxidized UQ to Qi site. We suggested that during sucrose gradient purification there is no complete oxidation and that is why there is some UQ6 bound in the Qp site. We made this statement, in the paper and will address it in our future studies. As to CL/PG and other PLs, despite a different method of purification, we found essentially all the same PLs and locations in the WT SC as well as the CRD1 Δ SC (except PG in CL sites). In our case the difference in the purification methods made our findings/conclusions even stronger.

p5, l-138. Typo in 'localizarion' and 'structue'.

(Corrected) We thank the reviewer for their corrections. Changes have been made. Line 185

p5 in 'Localizarion of Cardiolipin in the WT SC structue'. Are all the CL described here also in the previously resolved cryoEM SC structures? Please, clarify.

Yes, all cardiolipin (CL) in the WT were found in other structures. However, no cryo-EM structure identified all the CL in one single structure. Thus, we combined the CL observations from these studies and compared it to our experimental density map to complete the model. Inserted 'all' in line 190.

p8, l-248. 'Analysis of extracts from crd1 Δ mitochondria under more stringent conditions shows high levels of free CIII and CIV'. I cannot seem to find a gel in Ref 12 in support of that statement.

Western blot of BN-PAGE (Fig. 3 in Ref 13) shows exclusively free complexes in the mutant extracts. It should be noted that in these early experiments the blue dye was at 1% by weight while most investigators including us use about 0.1 to 0.3% in more recent reports, which may explain higher amounts of trimer (mostly) and some tetramer in mutant extracts.

p8, l-255. 'resulting in increased free individual complexes.' Again, ref 16 doesn't support more free CIII and CIV in the absence of CL.

Please see our new BN-PAGEs (Supplementary Fig. 1), and our answers above. In fact, previous ref 17 (Fig. 2A, center panel) shows an increase in trimer in the mutant and an increase in free CIV for the CL lacking strain. In the left panel there also appears to be detectible CIII in the CL lacking strain and none in the WT. However, no measurement of protein levels loaded on the different gels is listed so it is difficult to draw definite conclusions. More important, our new and previous data (as well as that of several others) support our statement of a difference between WT and CRD1 Δ strains regarding SC and individual complex distributions.

p12, l-379. 'Grids for cryo-EM were prepared as previously described (9).' Is this only true for the WT? Specify.

Now noted that previous method applies to the WT SC. Lines 425

p14, l-453. 'previously resolved yeast SCs and CIII (6YMX, 1KB9, 3CX5, 6Q9E, 6GIQ 11,16,18,41,42)'. Ref 18 is not from yeast.

Yes, Ref. 18 is not from yeast but ovine for 6Q9E. We used this structure for modeling since in this structure not only the 2 Qi sites are occupied by Ubiquinone but also there is a Ubiquinone in the Qo site. Line 158-160 retains Ref 18 but we have eliminated the source organism for each reference.

There is no mention of data deposition (EM maps and models) in appropriate data banks.

The deposition process has been completed and the EM maps and models will be available for public viewing upon publication release. The EMDB ids are EMD-27940 and EMD-28011 with their PDB ids being 8E7S and 8EC0 for the WT and mutant structures, respectively. Now noted under Data Deposition.

SI Table 2. What is being referred to as 'chains' in the table? 64 chains for the tetramer and 48 for the trimer? Please explain and clarify.

"Chains" was changed to "Ligands". The ligand numbers refer to the number of subunit-ligand interactions. This nomenclature was used in previous reports for the WT SC.

Reviewer #2 (Remarks to the Author):

The manuscript by Hryc et al describes the structure of the intact CIII2/CIV2 supercomplex from the yeast respiratory chain as compared to a CIII2/CIV1 complex isolated from a cardiolipin-deficient yeast mutant. The structure of the CIII2/CIV2 supercomplex revealed ubiquinone-6 molecules bound all four quinone binding sites and a series of cardiolipin molecules, in particular at the CIII/CIV interface. In the CIII2/CIV1 complex, fewer quinones were bound and the cardiolipin molecules were replaced by phosphatidylglycerol, which showed less interaction with the CIII and the CIV part of the complex.

The authors discuss their findings in relation to the role of cardiolipin for supercomplex stability and medical conditions involving impaired decrease cardiolipin levels and/or impaired respiratory supercomplex function.

These findings are interesting and will promote the field.

The authors should address the following comments:

1. The authors mention that the purified CIII2/CIV2 used for structure determination was enzymatically active, but refer to one of their previous papers. Please state activity values in the current manuscript for both CIII2/CIV2 and CIII2/CIV1.

Please see answer to Rev 1.

2. The authors argue that the stability of supercomplexes such as CIII2/CIV2 is reduced in absence of cardiolipin (e.g.: lines 68-72 and elsewhere in the manuscript). Did the authors experimentally confirm decreased stability for the purified CIII2/CIV1 complex, e.g.: by isothermal calorimetry, activity assays, etc? Does CIII2/CIV1 disintegrate into CIII and CIV over time (faster than CIII2/CIV2)?

Please see our introductory statement to reviewer comments and our response to Rev 1.

3. The phosphatidylglycerol content of the cardiolipin-deficient yeast mutant was determined by

mass spectrometry (lines 165-167). Please show the mass spectrometry result in an appropriate supplementary display item.

Mass spec results now in Supplementary Fig. 11.

4. The Qcr10 subunit apparently was lost during purification of CIII2/CIV1, but not of CIII2/CIV2. For the non-specialist reader it is difficult to assess which impact the presence or absence of this subunit may have on supercomplex stability. Please clearly indicate the location of Qcr10 in the CIII2/CIV2 structure.

Please see answer to Rev 1 for this question. Supplementary Fig. 8a shows Qcr10

Textual:

Line 53: "energy production", please rephrase

Line 54 now refers to line 59 where this is discussed further.

Line 123: "the" **Line 180**

Line 129: "red-ox" **Line 170**

Line 207: "this" **Line 253**

These have been corrected. Thank you.

Reviewers' Comments:

Reviewer #1:

Remarks to the Author:

Hryc et al have significantly improved their manuscript by addressing the reviewers' comments.

The following (mostly minor) points should be addressed:

1. l. 43-47, 'Although there is high sequence homology between yeast and 43 mammalian CIII and CIV, the latter SC contains complex I (CI, NADH:ubiquinone oxidoreductase) and is organized in linear array of CI-CIII-CIV. This arrangement is due to asymmetry within CIII (only one of the two monomers contains subunit Sub9). Sub9 in conjunction with assembly factor SCAF1 stabilizes interaction with CIV over that of CI'. This section makes no sense and should be revised; what is meant by 'and is organized in linear array of CI-CIII-CIV'? Please rewrite.
2. l. 100. I may be wrong but I don't think that in ref 10 and 16 the tag is on Cox5a. Please check.
3. l. 112. 'or CIV 11'. Two lines above it is written that in ref 11 the tag is on CIII? Please check.
4. More important. I disagree with the authors' interpretation of their new Western blots and their description of proportion of tetramer and trimer. Specifically, l. 116 'displayed almost exclusively the tetrameric SC'; it would be more accurate to write that they displayed 'predominantly' the tetrameric SC. Similarly l. 118, 'with trace amounts of the tetrameric SC' would be better replaced with 'significantly less tetrameric SC'.

Reviewer #2:

None

RESPONSE TO REVIEWERS' COMMENTS

Reviewer #1 (Remarks to the Author):

Hryc et al have significantly improved their manuscript by addressing the reviewers' comments.

Thank you for your positive response to our revised manuscript

The following (mostly minor) points should be addressed:

1. l. 43-47, 'Although there is high sequence homology between yeast and 43 mammalian CIII and CIV, the latter SC contains complex I (CI, NADH:ubiquinone oxidoreductase) and is organized in linear array of CI-CIII-CIV. This arrangement is due to asymmetry within CIII (only one of the two monomers contains subunit Sub9). Sub9 in conjunction with assembly factor SCAF1 stabilizes interaction with CIV over that of CI'. This section makes no sense and should be revised; what is meant by 'and is organized in linear array of CI-CIII-CIV'? Please rewrite.

This section been reworded as follows:

Although there is high sequence homology between yeast and mammalian CIII and CIV, the mammalian core respiratory SC organization differs from yeast in that it contains complex I (CI, NADH:ubiquinone oxidoreductase) and only one CIV¹². This difference in arrangement is due to asymmetry within CIII where only one of the two monomers contains subunit Sub9. Sub9 in conjunction with assembly factor SCAF1 stabilizes interaction of CIV over that with CI with one monomer of CIII. Lack of Sub9 in the other monomer allows interaction with CI resulting in CI and CIV flanking CIII.

2. l. 100. I may be wrong but I don't think that in ref 10 and 16 the tag is on Cox5a. Please check.

Reviewer is correct. The tag was on Cox13 and has been changed from Cox5a. The next comment (#3) is also correct and text has been changed as per below:

which were purified by affinity chromatography of a Qcr7-Flag-tagged^{11,17} or Cox13-His-tagged^{10,16} SC. More important the protein portion of our structure for the CRD1Δ SC trimer is also in close agreement with our WT SC and the published SCs purified using a tagged subunit of either CIII^{11,17} or CIV^{10,16}.

3. l. 112. 'or CIV 11'. Two lines above it is written that in ref 11 the tag is on CIII? Please check.

4. More important. I disagree with the authors' interpretation of their new Western blots and their description of proportion of tetramer and trimer. Specifically, l. 116 'displayed almost exclusively the tetrameric SC'; it would be more accurate to write that they displayed 'predominantly' the tetrameric SC. Similarly l. 118, 'with trace amounts of the tetrameric SC' would be better replaced with 'significantly less tetrameric SC'.

Changed as per reviewer's suggestion:

We verified previous reports that digitonin extracts of WT mitochondria¹³ subjected to BN-PAGE displayed predominately the tetrameric SC with trace amounts of the trimeric SC or free

CIII and CIV (See Supplementary Fig. 1). Extracts from *CRD1*Δ mitochondria showed large amounts of the trimeric SC as well as free CIII and CIV with significantly lesser amounts of the tetrameric SC confirming previous results

Reviewer #2 (Remarks to the Author):

The authors have addressed my concerns. In my view this manuscript can be published.

No concerns raised